# PLA: A Principled Path from Softmax Attention to Linear Models via KV Cache Compression

## Abstract

Transformers, despite their remarkable sequence modeling capabilities, are fundamentally constrained by the quadratic complexity of Softmax attention and the unbounded growth of the key–value (KV) cache. Replacing Softmax attention with linear variants has emerged as a promising direction, yet existing approaches lack a systematic functional comparison with Softmax attention, clear error analysis, and a theoretically guided roadmap for improvement. In this work, we approach the problem from the perspective of KV cache compression and present a theoretically grounded pathway from Softmax attention to linear models. Our analysis reveals five critical components: redundancy elimination, tokenizer-level quantization and positional information separation, positional information compression, inter-layer similarity, and multi-state decomposition. For each, we provide succinct theoretical justification, derive error bounds, and demonstrate equivalence to existing mechanisms. Building on this pathway, we introduce PLA, a linearized attention model that inherits pretrained weights and achieves state-of-the-art performance. Notably, PLA surpasses strong baselines such as MVA and GSA on multiple benchmarks while requiring only 80% of the fine-tuning resources. Our findings provide both theoretical clarity and practical guidance for advancing linear attention, highlighting a principled route towards efficient and scalable alternatives to Softmax attention.

## 1 Introduction

The Transformer architecture Vaswani et al. (2017) has become the backbone of modern deep learning, powering state-of-the-art models in language Touvron et al. (2023); Jiang et al. (2023); DeepSeek-AI et al. (2024); Dubey et al. (2024); Yang et al. (2025), vision Dosovitskiy et al. (2020); Han et al. (2022), and multimodal Yin et al. (2024) domains due to its remarkable sequence modeling capability. Despite these successes, Transformers face two fundamental limitations: the *quadratic complexity* of Softmax attention with respect to sequence length, and the *unbounded growth* of the key-value (KV) cache during autoregressive inference. These issues severely constrain the applicability of Transformers in long-sequence modeling tasks, such as video understanding Tang et al. (2025), genomic sequence analysis Jumper et al. (2021), and other domains requiring extended context Jiang et al. (2025).

To address these limitations, two major lines of research have emerged: *KV cache compression* Luohe et al. (2024); WEI et al. (2025) and *linear attention* Katharopoulos et al. (2020); Hua et al. (2022); Qin et al. (2024a) Lingle (2024) Alonso et al. (2026). KV cache compression methods aim to reduce memory usage by compressing the stored states across either the sequence or the channel dimension. For example, some approaches design task-specific prompts that select a fixed set of relevant KV entries to retain, thus improving efficiency at the cost of generality. Others Hu et al. (2021) apply low-rank or dimensionality reduction techniques along the channel dimension, yielding constant compression ratios but failing to fundamentally address the ever-growing cache size in long contexts.

Linear attention methods, in contrast, replace the Softmax kernel with kernelized approximations, thereby reordering computations to achieve linear complexity. Crucially, such models can recursively maintain a fixed-size state, resolving the KV cache growth problem. Recent works Yang et al. (2024b;a) further en-

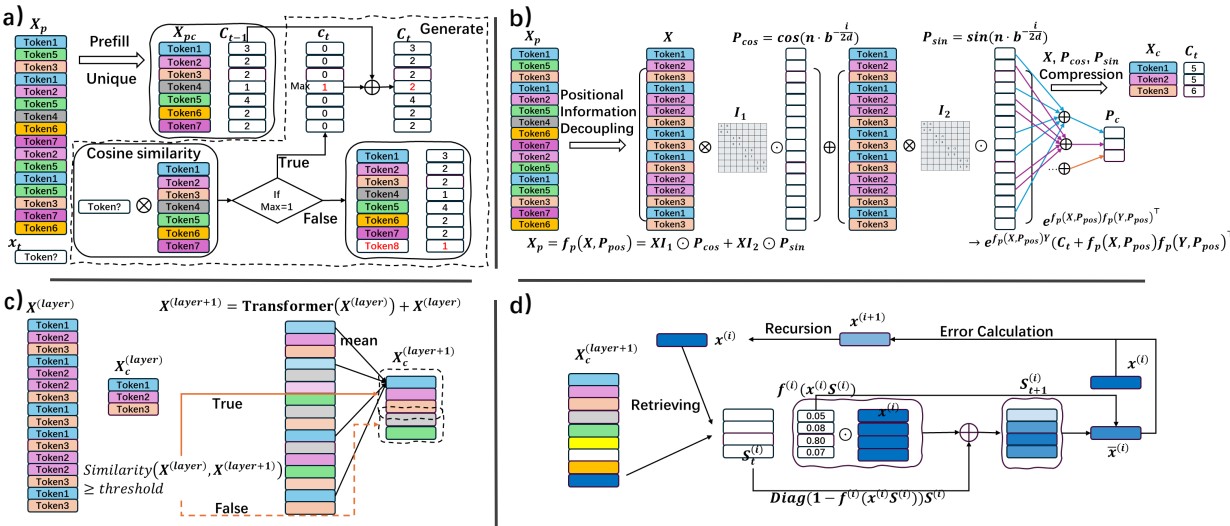

Figure 1: **Conceptual illustration of the RSA memory hierarchy and its theoretical foundations.**
This figure illustrates the key components that enable bounded state size while preserving representational capacity, grounded in the theoretical analysis of Section 3. **(a) Redundancy elimination.** During sequence growth, a large number of repeated tokens appear. The unbounded growth of the KV cache can be replaced by storing an identical token and recording its occurrence count. When the cosine similarity between tokens is greater than or equal to a threshold (e.g., 1 or 0.95), the two tokens are considered identical; the count is incremented. Otherwise, the new token is appended to the KV cache. This mechanism transforms the unbounded KV cache into a KV state with a bounded storage capacity. **(b) Position decoupling and position compression.** After decoupling positional information from the sequence, the number of token types is significantly reduced. For instance, after decoupling positional information at the first layer, the number of token types equals the vocabulary size of the tokenizer. The positional information is then linearly compressed and used to compute attention scores (approximating a first-order Taylor expansion). **(c) Inter-layer similarity.** Although the number of KV token types in subsequent layers increases substantially due to token mixing operations, it remains bounded by the previous layer's capacity due to the residual network structure. Consequently, the upper bound on the KV cache in subsequent layers can be controlled via the first-layer tokenizer vocabulary size. Specifically, if the cosine similarity between a token in a later layer and its corresponding token in the previous layer is below a threshold, it is compressed into a fixed state; otherwise, it is stored using a state size equal to that of the previous layer. **(d) Multi-level state decomposition.** The upper bound imposed by the tokenizer vocabulary size offers limited advantage compared to current task-specific KV cache compression methods. We therefore approximate this bound using a multi-level decomposition. For example, a vocabulary size of $128K$ can be approximated by a three-level linear state with size 64 per level (since $64^3 > 128K$). The input feature $x^{(i)}$ is distributed across the states $S^{(i)}$ via retrieval scores; the information of $x^{(i)}$ in $S^{(i)}$ is then retrieved and reconstructed as $\bar{x}^{(i)}$. The residual error between $x^{(i)}$ and $\bar{x}^{(i)}$ is computed as $x^{(i+1)}$ and stored recursively.

hance these models with additional mechanisms such as gating functions and delta-rule updates to improve expressiveness. However, linear attention models still exhibit significant drawbacks: they often suffer from limited retrieval and reasoning capacity, exhibit noticeable performance gaps relative to Softmax attention, and typically require training from scratch or hybridization with Softmax attention to achieve competitive results.

In this work, we revisit the connection between Softmax attention and linear attention from the perspective of *KV cache compression*. We propose what we argue to be the current optimal and theoretically grounded pathway for compressing Softmax attention into linear models. This pathway is structured around five theoretical principles, each demonstrating (i) the necessity of a specific compression step, (ii) its equivalence to mechanisms in existing approaches, and (iii) the error it introduces relative to Softmax attention. Taken

together, these principles provide a clear functional blueprint of what linear attention should retain, what it can safely discard, and how it differs fundamentally from Softmax attention.

Our analysis yields both theoretical and practical benefits. First, it clarifies the essential components required to bridge the gap between Softmax and linear attention, guiding future designs of efficient architectures. Second, it enables the transformation of pretrained Softmax-based large language models into linear variants with significantly reduced fine-tuning cost. Empirically, we show that our approach achieves state-of-the-art performance, narrowing the gap between linear and Softmax attention especially on tasks where existing linear models struggle, such as retrieval, few-shot reasoning, and complex logical inference.

## 2 Background and Preliminaries

### 2.1 Transformers and Softmax Attention

The Transformer architecture relies on the attention mechanism to dynamically compute contextualized representations. Given an input sequence $\boldsymbol{X} \in \mathbb{R}^{t \times d}$, it is linearly projected into queries $\boldsymbol{Q}$, keys $\boldsymbol{K}$, and values $\boldsymbol{V}$. Attention is then computed as

$$\boldsymbol{O} = \text{Attention}(\boldsymbol{Q}, \boldsymbol{K}, \boldsymbol{V}) = \text{Softmax}\left(\frac{\boldsymbol{Q}\boldsymbol{K}^\top}{\sqrt{d}} \odot \boldsymbol{M}\right)\boldsymbol{V}, \boldsymbol{o}_i = \sum_{j=1}^{i} \frac{\exp\left(\frac{\boldsymbol{q}_i \boldsymbol{k}_j^\top}{\sqrt{d}}\right)}{\sum_{h=1}^{i} \exp\left(\frac{\boldsymbol{q}_i \boldsymbol{k}_h^\top}{\sqrt{d}}\right)} \boldsymbol{v}_j, \tag{1}$$

where $\boldsymbol{M}$ denotes the causal mask with $\boldsymbol{M}_{ij} = 1$ if $i \geq j$ o.w. $-\infty$ and $d$ is the feature dimension used for normalization. Equivalently, the autoregressive form can be written as $\boldsymbol{o}_i$, where $\boldsymbol{q}_i$, $\boldsymbol{k}_j$, and $\boldsymbol{v}_j$ are the $i$-th or $j$-th row vectors of $\boldsymbol{Q}$, $\boldsymbol{K}$, and $\boldsymbol{V}$, respectively. While highly effective, this formulation entails quadratic complexity in sequence length and requires storing all past key–value pairs, leading to unbounded KV cache growth during inference.

### 2.2 KV Cache Compression Methods

To alleviate the quadratic growth of the key–value (KV) cache, a large body of work explores *KV cache compression*. The central idea is to reduce redundancy in the cache by performing low-rank transformations or selection operations along the sequence dimension or the channel dimension.

Formally, given an input $\boldsymbol{X} \in \mathbb{R}^{t \times d}$ and its projections $\boldsymbol{Q}, \boldsymbol{K}, \boldsymbol{V} \in \mathbb{R}^{t \times d}$, the compressed cache $(\boldsymbol{K}^c, \boldsymbol{V}^c)$ of size $c \times d^r$ is defined as

$$\boldsymbol{K}^c = \varphi\big(\phi(\boldsymbol{R}\boldsymbol{K})\boldsymbol{L}\big), \qquad \boldsymbol{V}^c = \varphi\big(\phi(\boldsymbol{R}\boldsymbol{V})\boldsymbol{L}\big), \tag{2}$$

where $\boldsymbol{R} \in \mathbb{R}^{c \times t}$ selects $c$ tokens from the sequence, and $\boldsymbol{L} \in \mathbb{R}^{d \times d^r}$ compresses the channel dimension. Here $d^r = h \times d_k^r$, with $h$ denoting the number of heads and $d_k^r$ the per-head compressed dimension. The operators $\phi(\cdot)$ and $\varphi(\cdot)$ denote transformation and selection functions, respectively.

Because the sequence length $t$ grows without bound during autoregressive generation, $\boldsymbol{R}$ is usually constructed recursively, i.e.,

$$\boldsymbol{R} = f(\boldsymbol{R}'\boldsymbol{X}^\top) \in \mathbb{R}^{c \times t}, \tag{3}$$

where $\boldsymbol{R}' \in \mathbb{R}^{c \times d}$ defines a local observation window, and $f(\cdot)$ specifies the selection strategy.

(1) **SnapKV** Li et al. (2025) applies compression only along the sequence dimension. Specifically, $\boldsymbol{L}$ is the identity matrix, so no channel compression is applied. $\boldsymbol{R} = f(\boldsymbol{Q}\boldsymbol{K}^\top)$ is defined via a top-$c$ operator over the most recent queries and obtains the indexes of the corresponding KV block. $\phi(\cdot)$ gather the corresponding key tokens by these indices. The same procedure is applied to $\boldsymbol{V}$. (2) **HeadKV** Fu et al. (2025) extends SnapKV by compressing along the channel dimension. In this case, $\boldsymbol{R}$ again selects tokens as in SnapKV, while $\boldsymbol{L}$ is defined through an additional projection $\boldsymbol{L}'$, which serves as a voting mechanism across attention heads. $\varphi(\cdot)$ gather the corresponding key heads by these votes. (3) **Multi-Head Latent Attention (MLA)** DeepSeek-AI et al. (2025) approach removes explicit token selection. Both $\phi(\cdot)$ and $\varphi(\cdot)$ are set to the identity. No sequence compression is applied, i.e., $\boldsymbol{R} = \boldsymbol{I}$. Instead, channel compression is performed

with a fixed projection $\boldsymbol{L} = (\boldsymbol{W}^{UK})^{-1}$. By leveraging the associativity of matrix multiplication to fuse the up-projection with the query-key product, MLA avoids explicitly reconstructing $\boldsymbol{K}$ and $\boldsymbol{Q}$, achieving substantial memory savings and computational speedup.

Numerous subsequent approaches, including InfLLM Xiao et al. (2024a), HO2 Zhang et al. (2023), and StreamLLM Xiao et al. (2024b), can be understood as hybrids or equivalent reformulations of the above principles.

## 2.3 Linear Models

Linear attention replaces the Softmax kernel with linearizable feature maps, which permits re-ordering the computations among queries, keys and values and thereby achieves linear time and fixed-size state:

**Parallel form.**
$$\boldsymbol{O} = \text{LA}\big(\phi(\boldsymbol{Q}), \phi(\boldsymbol{K}), \boldsymbol{V}\big) = \big((\phi(\boldsymbol{Q})\phi(\boldsymbol{K})^\top) \odot \boldsymbol{M}\big)\boldsymbol{V}, \tag{4}$$

**Recursive form.**
$$\boldsymbol{S}_t = \boldsymbol{S}_{t-1} + \phi(\boldsymbol{k}_t)^\top \boldsymbol{v}_t, \qquad \boldsymbol{o}_t = \phi(\boldsymbol{q}_t)\,\boldsymbol{S}_t, \tag{5}$$

where $\boldsymbol{S}_t \in \mathbb{R}^{d_k \times d_v}$ is a fixed-size state matrix maintained across time steps, $t$ is a numerical value or index for the time dimension. By keeping $\boldsymbol{S}_t$ bounded, linear attention attains constant memory during autoregressive inference. Many works focus on improving the choice of $\phi(\cdot)$ Han et al. (2023); Choromanski et al. (2022) or introducing auxiliary mechanisms to enhance expressiveness.

However, this formulation is prone to state saturation, which dilutes the attention mechanism. To address this, methods like GLA introduce a gating mechanism that enables dynamic forgetting in the state $\boldsymbol{S}_t$, thereby promoting a bias towards more recent context.

**Gating / GLA.** Gated Linear Attention (GLA) Yang et al. (2024b) applies multiplicative gates to control the contribution of new tokens and the persistence of prior state:

$$\boldsymbol{O} = \text{GLA}(\boldsymbol{Q}, \boldsymbol{K}, \boldsymbol{V}, \boldsymbol{G}) = \text{LA}(\boldsymbol{Q} \odot \boldsymbol{B}, \ \boldsymbol{K}/\boldsymbol{B}, \ \boldsymbol{V}), \boldsymbol{S}_t = \text{diag}(\boldsymbol{g}_t)\,\boldsymbol{S}_{t-1} + \boldsymbol{k}_t^\top \boldsymbol{v}_t, \boldsymbol{o}_t = \boldsymbol{q}_t\,\boldsymbol{S}_t. \tag{6}$$

where the $t$-th row of $\boldsymbol{B}$ is $\boldsymbol{b}_t = \prod_{i=1}^t \boldsymbol{g}_i$ and $\boldsymbol{G} = \sigma(\boldsymbol{X}\boldsymbol{W}_g) \in \mathbb{R}^{n \times d_k}$. While models like MetaLA Chou et al. (2024), HGRN2 Qin et al. (2024b), and GSA Zhang et al. (2024) also employ gating, this approach offers a relatively coarse control over the state $S_t$, failing to fully address information redundancy. This limitation motivated the development of more sophisticated updates, such as those inspired by fast weights.

**Fast-weight / Delta Rule-style updates.** These methods aim to correct the stored representation based on prediction error:

$$\boldsymbol{v}_t^{\text{old}} = \boldsymbol{k}_t\,\boldsymbol{S}_{t-1}, \qquad \boldsymbol{S}_t = \boldsymbol{S}_{t-1} + \boldsymbol{g}_t \cdot \boldsymbol{k}_t^\top\big(\boldsymbol{v}_t - \boldsymbol{v}_t^{\text{old}}\big), \qquad \boldsymbol{o}_t = \boldsymbol{q}_t\,\boldsymbol{S}_t. \tag{7}$$

Intuitively, the update adds a correction proportional to the discrepancy between the newly observed value $\boldsymbol{v}_t$ and the state-predicted value $\boldsymbol{v}_t^{\text{old}}$, gated by $\boldsymbol{g}_t$. Building on this, DeltaNet Yang et al. (2024a) parallelized the formulation to create a powerful linear model.

**Optimization viewpoint.** Furthermore, viewing the update through an optimization lens led to dynamic weighting methods like TTT Sun et al. (2025), Titans Behrouz et al. (2024), and Atlas Agrawal et al. (2025), which share the unified objective:

$$\mathcal{L}_t(\boldsymbol{M}) \ = \ \sum_{i=1}^t \gamma_i \left\| \boldsymbol{M}\big(\phi(\boldsymbol{k}_i)\big) - \boldsymbol{v}_i \right\|_2^2, \tag{8}$$

where $\boldsymbol{M}(\cdot)$ denotes a (possibly parametric) mapping from key-features to value-predictions and $\{\gamma_i\}$ are weighting coefficients. One may then update the memory/map $\boldsymbol{M}_t$ by performing a few steps of iterative optimization (e.g., gradient descent-GD or Muon Jordan et al. (2024)-style). A compact schematic of such an optimization-inspired update is:

$$\boldsymbol{M}_t = \alpha_t\,\boldsymbol{M}_{t-1} + F(\boldsymbol{S}_t), \quad \boldsymbol{S}_t = \eta_t\,\boldsymbol{S}_{t-1} - \theta_t\,\nabla_{\boldsymbol{S}}\,\mathcal{L}_t\big(\boldsymbol{M}_{t-1}; \boldsymbol{k}_t, \boldsymbol{v}_t\big), \tag{9}$$

where $F(\cdot)$ aggregates the current statistics into the primary memory $\boldsymbol{M}_t$, and the second line denotes a gradient-based (or similar) corrective step for the working state $\boldsymbol{S}_t$. This optimization viewpoint explains a number of empirically successful update rules and motivates algorithms that explicitly minimize per-step predictive error.

**Extending the state budget.** Even with sophisticated updates, a single low-rank state may remain insufficient to capture complex, multi-scale dependencies. Recent works therefore maintain multiple parallel or hierarchical states, each specialized for different temporal ranges or functional roles. For instance, **MoM** Du et al. (2025) and **MVA** Wang et al. (2025) maintain multiple memory banks (e.g., short-term vs. long-term) and/or decompose the state into several sub-states that interact during read/write.

$$\boldsymbol{S}_t^{(i)} = \mathrm{diag}(1 - \bar{f}^{(i)}(\boldsymbol{x}_t^{(i)})^\top)\boldsymbol{S}_{t-1}^{(i)} + \bar{f}^{(i)}(\boldsymbol{x}_t^{(i)})^\top \boldsymbol{x}_t^{(i)}, \qquad \boldsymbol{x}_t^{(i+1)} = f_1^{(i)}(\boldsymbol{x}_t^{(i)}, \boldsymbol{S}_t^{(i)}) \tag{10}$$

where $f$ function is generally taken as $\sigma$, while the $f_1$ function is taken as a hybrid expert or delta function. This multi-state design can close much of the performance gap to Softmax attention, at the cost of additional architectural and algorithmic complexity.

## 3 Method

Existing KV cache compression methods either lack general applicability or fail to address the unbounded growth of KV cache, while also lacking a clear error analysis compared to Softmax Attention. Furthermore, current linear attention approaches lack a comprehensive understanding of their components and mechanisms from the perspective of Softmax Attention, resulting in the absence of clear improvement strategies to match or even surpass Softmax Attention performance.

To address these limitations, this paper presents five theoretical principles with corresponding experimental validation, establishing an optimal pathway for compressing Softmax Attention into linear attention. Each theoretical node provides rigorous error analysis and demonstrates equivalence to existing model operations and mechanisms, thereby offering valuable references and guidance for future improvement strategies.

### 3.1 Necessity of Redundancy Removal

**Proposition 1** (Necessity of Deduplication and Storage Upper Bound). *Let each component of the key vectors $\mathbf{k}_t \in \mathbb{R}^{d_k}$ and value vectors $\mathbf{v}_t \in \mathbb{R}^{d_v}$ be stored with $b$ bits (e.g., $b = 16$ for `float16`, $b = 4$ for `int4`). Consider an arbitrary finite or infinite sequence $\{(\mathbf{k}_t, \mathbf{v}_t)\}_{t=1}^T$ (where $T$ can be infinite). Define the set of all distinct key-value pairs appearing in the sequence as*

$$\mathcal{U} = \left\{ (\mathbf{u}_i, c_i) \right\}_{i=1}^C,$$

*where $\mathbf{u}_i = [\mathbf{k}_{p_i}; \mathbf{v}_{p_i}] \in \mathbb{R}^{d_k + d_v}$ is the concatenation of a key-value pair that has occurred, and $c_i \in \mathbb{N}$ denotes the number of times this pair appears in the entire sequence. Then the following conclusions hold:*

1. ***Upper Bound on Storage** $C$: The number of distinct key-value pairs satisfies*

$$C \leq 2^{b(d_k + d_v)}.$$

*If each count $c_i$ is stored with $b_c$ bits (e.g., $b_c = \lceil \log_2 T \rceil$ suffices for the entire sequence), then the total number of bits $S_b$ required to store the compressed representation is bounded by*

$$S_b = C \times \left( b(d_k + d_v) + b_c \right) \leq 2^{b(d_k + d_v)} \times \left( b(d_k + d_v) + b_c \right).$$

2. ***Losslessness (Attention Equivalence):** For any query vector $\mathbf{q} \in \mathbb{R}^{d_k}$, the attention output computed using the original sequence is identical to that computed using the compressed representation of unique key-value pairs with their counts, i.e.,*

$$\sum_{t=1}^T \frac{\exp(\mathbf{q}^\top \mathbf{k}_t)}{\sum_{s=1}^T \exp(\mathbf{q}^\top \mathbf{k}_s)} \mathbf{v}_t = \sum_{i=1}^C c_i \frac{\exp(\mathbf{q}^\top \mathbf{k}_i)}{\sum_{j=1}^C c_j \exp(\mathbf{q}^\top \mathbf{k}_j)} \mathbf{v}_i.$$

3. **Connection to the Delta Rule in Fast Weight Models**: *If each unique key-value pair is viewed as an associative memory unit, the update rule of the above deduplication-with-counting mechanism is equivalent to a Delta Rule capable of independently storing new information (see Corollary A.1.3 in the appendix for details).*

A detailed proof is provided in the Appendix A.1.

**Generalization.** The unique operation can also be generalized by relaxing exact matching to cosine similarity. For example, using a threshold of 0.9 instead of 1.0 can still preserve performance in A.1. This opens the possibility of balancing efficiency and accuracy by tuning the threshold. Furthermore, we propose a more general and stronger compression approach: when two tokens exhibit cosine similarity above a threshold, we treat them as identical and replace them by their average. This can be interpreted as a quantization process, which also provides noise reduction.

**Complexity Implication.** Let the upper bound of the KV cache be $C$. According to the above reasoning, Softmax attention can be interpreted as a linear model with complexity $\mathcal{O}(N \times C \times d)$, where $C = 2^{2bd}$. Since $C$ is extremely large, the naive bound is impractical in comparison to specialized task-optimized methods. Nonetheless, the key insight here is that redundancy removal (via unique or Delta Rule operations) is *necessary* to compress an unbounded state into a bounded one.

### 3.2 Tokenization and Positional Information Decoupling

To achieve stronger compression, we conduct a deeper analysis of sequence modeling in LLMs. The input to an LLM undergoes tokenization, which constitutes a strong quantization that limits the number of distinct types to the vocabulary size $V_T$ (e.g., 32K for LLaMA2). Subsequent channel mixing operations in the LLM do not affect the number of distinct types; rather, it is the positional encoding and token mixing operations that impact type diversity. We therefore optimize the input-output characteristics of these two operations, leveraging the first-layer tokenization to achieve stronger compression and reduce the upper bound of the state storage requirement.

We introduce two optimizations specifically targeting positional encoding operations:

**Proposition 2** (Necessity of Positional Information Decoupling)**.** *Storage Bound for Positional Decoupling in the First Layer: Let the vocabulary size be $V \in \mathbb{N}$, and assume each token corresponds to a unique word embedding vector $\boldsymbol{e}_v \in \mathbb{R}^d$ ($v = 1, \ldots, V$). Denote the embedding matrix as $\boldsymbol{E}_{vocab} = [\boldsymbol{e}_1, \ldots, \boldsymbol{e}_V]^\top \in \mathbb{R}^{V \times d}$. Consider an input sequence of length $T$ with token indices $w_1, \ldots, w_T \in \{1, \ldots, V\}$. Suppose the model employs a positional encoding function*

$$PE : \mathbb{R}^{T \times d} \times \mathbb{R}^{T \times d} \to \mathbb{R}^{T \times d},$$

*which combines the word embedding sequence $\boldsymbol{E}_{seq} = [\boldsymbol{e}_{w_1}, \ldots, \boldsymbol{e}_{w_T}]^\top \in \mathbb{R}^{T \times d}$ with a positional encoding matrix $\boldsymbol{P}(\boldsymbol{I}) \in \mathbb{R}^{T \times d}$ to produce the first-layer input*

$$\tilde{\boldsymbol{X}}^{(1)} = PE(\boldsymbol{E}_{seq}, \boldsymbol{P}(\boldsymbol{I})) \in \mathbb{R}^{T \times d}.$$

*Then there exists a lossless compression method that represents $\tilde{\boldsymbol{X}}^{(1)}$ as a triple $(\boldsymbol{E}, \boldsymbol{I}, \tilde{\boldsymbol{P}})$, where:*

- *$\boldsymbol{E} = \boldsymbol{E}_{vocab}$ (the vocabulary embedding matrix, which needs no extra storage);*

- *$\boldsymbol{I} \in \mathbb{N}^T$ is an index vector satisfying $\boldsymbol{I}_t = w_t$ (i.e. it directly stores the token index of each position);*

- *$\tilde{\boldsymbol{P}} \in \mathbb{R}^{T' \times d}$ is a positional information matrix obtained after decoupling $\boldsymbol{I}$ and $\boldsymbol{E}$, where $T'$ is a constant or equal to $T$ depending on the specific method, and we have*

$$\tilde{\boldsymbol{X}}^{(1)} = PE\big(\boldsymbol{E}[\boldsymbol{I}, :], \ \boldsymbol{P}(\tilde{\boldsymbol{P}}, \boldsymbol{I})\big),$$

*with $\boldsymbol{E}[\boldsymbol{I}, :] \in \mathbb{R}^{T \times d}$ denoting the matrix formed by selecting rows from $\boldsymbol{E}$ according to $\boldsymbol{I}$, and $\boldsymbol{P}$ is obtained from $\tilde{\boldsymbol{P}}$ and $\boldsymbol{I}$ through some function. For instance, with RoPE we have $T' = 1$.*

*Furthermore, this compression method enjoys the following properties:*

1. **Upper bound on the number of distinct row vectors:** *Let $\mathcal{U} = \{\tilde{\boldsymbol{x}}_t^{(1)} \mid t = 1, \ldots, T\}$ be the set of all distinct row vectors of $\tilde{\boldsymbol{X}}^{(1)}$. Then*

$$|\mathcal{U}| \leq V.$$

2. **Storage size:** *If each numerical component is stored with b bits (e.g., $b = 16$ for `float16`, $b = 4$ for `int4`), the total number of bits required for the compressed representation is*

$$\underbrace{V \cdot d \cdot b}_{\text{vocabulary embeddings (shareable)}} + \underbrace{T \cdot \lceil \log_2 V \rceil}_{\text{indices}} + \underbrace{T \cdot d \cdot b}_{\text{positional information } \tilde{\boldsymbol{P}}}.$$

*In general, after decoupling $\boldsymbol{P}$ into $\tilde{\boldsymbol{P}}$ and $\boldsymbol{I}$, the positional information can be heavily compressed. For the case of RoPE in large language models, the total storage required for the compressed representation becomes*

$$\underbrace{V \cdot d \cdot b}_{\text{vocabulary embeddings (shareable)}} + \underbrace{T \cdot \lceil \log_2 V \rceil}_{\text{indices}} + \underbrace{1 \cdot d \cdot b}_{\text{positional information } \tilde{\boldsymbol{P}}},$$

*where $\tilde{\boldsymbol{P}}$ is a predetermined row vector.*

Due to positional constraints, the upper bound remains identical to Proposition 1. We subsequently address this limitation through positional information compression. The detailed proof is provided in Appendix A.2. This approach provides lossless compression for the first layer.

### 3.3   Positional Information Compression

Building upon the upper bound established in Proposition 2, we further optimize the positional information representation to achieve the same upper bound as the position-agnostic KV cache in Proposition 2. This allows us to completely control the first-layer upper bound at the vocabulary size level, laying the foundation for fully fixed-size state linear models.

**Proposition 3** (Necessity of Positional Information Compression)**.** *Assume the model uses RoPE (Rotary Position Embedding) with head dimension d (even). Define the rotation angles*

$$\theta_j = \text{base}^{-\frac{2j}{d}}, \quad j = 0, 1, \ldots, \frac{d}{2} - 1,$$

*where $\text{base} > 1$ is a fixed base parameter. For any position index m, let the compressed positional information be stored as a linear superposition*

$$p_m^{(t)} = p_{m1} + p_{m2} + \cdots + p_{mt},$$

*which represents the accumulated effect of positions up to some reference. The positional encoding function $f_p(\boldsymbol{x}_m, \boldsymbol{p}(m))$ combines a vector $\boldsymbol{x}_m$ with its positional code $\boldsymbol{p}(m)$; for RoPE it is defined as*

$$f_p(\boldsymbol{x}_m, \boldsymbol{p}(m)) = \begin{bmatrix} x_0 & x_1 & x_2 & x_3 & \ldots & x_{d-1} & x_d \end{bmatrix} \odot \boldsymbol{p}^c(m)$$
$$+ \begin{bmatrix} x_1 & -x_0 & x_3 & -x_2 & \ldots & x_d & -x_{d-1} \end{bmatrix} \odot \boldsymbol{p}^s(m),$$

*with*

$$\boldsymbol{p}^c(m) = \begin{bmatrix} \cos(m\theta_0) & \cos(m\theta_0) & \ldots & \cos(m\theta_{d/2-1}) & \cos(m\theta_{d/2-1}) \end{bmatrix},$$
$$\boldsymbol{p}^s(m) = \begin{bmatrix} \sin(m\theta_0) & \sin(m\theta_0) & \ldots & \sin(m\theta_{d/2-1}) & \sin(m\theta_{d/2-1}) \end{bmatrix}.$$

*Consider the following approximation to the original attention score:*

$$\tilde{A}_{n,s} = e^{f_p(\boldsymbol{q}_n, \boldsymbol{p}(n))^\top \boldsymbol{k}_s} \left( t + f_p(\boldsymbol{q}_n, \boldsymbol{p}(n))^\top f_p(\boldsymbol{k}_s, \boldsymbol{p}^{(t)}(s)) - f_p(\boldsymbol{q}_n, \boldsymbol{p}(n))^\top \boldsymbol{k}_s \right),$$

where $t$ is a scalar (e.g., count) and $\boldsymbol{p}^{(t)}(s)$ denotes the compressed positional information up to step $t$. Under the assumption that base is sufficiently large so that for all relevant relative positions $\Delta = n - s$ we have $|\Delta|\theta_j \leq 1$ for every $j \geq 2$, the approximation error satisfies

$$|A_{n,s} - \tilde{A}_{n,s}|_j = O\left(\frac{n}{\text{base}^{4j/d}}\right).$$

Thus, when base is large, $\tilde{A}_{n,s}$ closely approximates the exact attention, with an error that decays as base increases. For small $j$ the error may be larger, but it is diluted by the normalization over dimension $d$.

The detailed proof is provided in Appendix A.3.

The separation of positional information effectively emulates gating mechanisms used in state-of-the-art architectures. Theoretical results show that this compression is most effective in the **first layer**. In deeper layers, where token representations are increasingly entangled with positional cues, simple redundancy reduction cannot maintain the vocabulary-size bound. To address this, we introduce a **layer-wise similarity constraint**, where the **expected cosine similarity** between adjacent hidden states serves as a **contractive factor**. This yields a recursive bound that restricts the memory footprint of intermediate layers to a **constant multiple of the vocabulary size**, ensuring a fixed-state regime while preserving expressivity.

### 3.4 Inter-layer Similarity and State Propagation

**Proposition 4** (Inter-layer Similarity and Recursive Compression Upper Bound). *Consider a Transformer model with residual connections. Let the input to layer $l$ be $\boldsymbol{X}^{(l)} \in \mathbb{R}^{T \times d}$, where $T$ is the sequence length and $d$ is the dimension. Define $\mathcal{U}_l = \{\boldsymbol{u}_i^{(l)}\}_{i=1}^{M_l}$ as the set of all distinct row vectors of $\boldsymbol{X}^{(l)}$, and let $M_l = |\mathcal{U}_l|$ denote the number of unique vectors in that layer. Suppose the numerical precision is such that the Euclidean distance between any two distinct vectors is at least $\varepsilon > 0$ (i.e., $\|\boldsymbol{u} - \boldsymbol{v}\| \geq \varepsilon$ for $\boldsymbol{u} \neq \boldsymbol{v}$). Define the threshold $\tau = 1 - \varepsilon^2/8$; then any two distinct vectors cannot simultaneously have cosine similarity $\geq \tau$ with the same vector. (Indeed, if $\|\boldsymbol{u} - \boldsymbol{c}\| \leq r$ and $\|\boldsymbol{v} - \boldsymbol{c}\| \leq r$ with $r = \sqrt{2(1-\tau)}$, then $\|\boldsymbol{u} - \boldsymbol{v}\| \leq 2r < \varepsilon$, a contradiction.)*

*For layer $l$ and layer $l-1$, define the quantization function*

$$Q_{ij} = \begin{cases} 1 & \text{if } \cos(\boldsymbol{u}_i^{(l)}, \boldsymbol{u}_j^{(l-1)}) \geq \tau, \\ 0 & \text{otherwise.} \end{cases}$$

*Let*

$$\mathbb{E}[Q(X^{(l)}, X^{(l-1)})] = p_l = \frac{1}{M_l} \sum_{i=1}^{M_l} \max_j Q_{ij},$$

*i.e., the proportion of unique vectors in layer $l$ that achieve the threshold similarity with some vector in layer $l-1$. Then the following upper bound holds:*

$$M_l \leq \min\left(\frac{M_{l-1}}{p_l}, T\right).$$

*Analogously, by symmetry we also have*

$$M_{l-1} \leq \min\left(\frac{M_l}{p_l'}, T\right),$$

*where $p_l'$ is the reverse proportion (i.e., the proportion of unique vectors in layer $l-1$ that achieve the threshold similarity with some vector in layer $l$). And $M_0$ is bounded by the vocabulary size, $T$ is the original sequence length, the upper bound $M_L$ for the final layer also approaches the vocabulary size due to the vocabulary projection.*

The detailed proof is provided in Appendix A.4. Although the analysis is simplified, empirical results confirm its general applicability. For example, when using inputs from selected layers [0,1,2,5,8,11,14,15,17,18,19,22]

as subsequent layer inputs, Mistral-7B achieves 100% accuracy on the passkey retrieval task. This effect arises from inter-layer similarity, where residual connections propagate redundancy reduction while preserving information.

Since the similarity lower bound may approach zero—leading to intermediate states close to the original sequence length—we introduce a constant scaling factor $c \in [1, 2)$ for practical control. Specifically, the state size of layer $i$ is set as $(c - 1)N_{i-1}$. Empirically: 1. Beyond certain lengths, attention between new queries and stored states becomes sparse (e.g., NSA, MoBA). 2. Many tasks succeed with fixed-size states (e.g., SnapKV, GLA, GSA). 3. Inference typically operates within bounded state spaces. However, for large-vocabulary models (e.g., Qwen2.5, LLaMA3 with $\sim$128K tokens), even moderate scaling (e.g., $c = 1.5$ in a 32-layer model) leads to $\sim$ 82M states, necessitating further compression. Inspired by MVA's vocabulary decomposition and MoM's functional partitioning, we adopt multi-memory states to approximate Proposition 4 bounds while preserving fixed-size representations.

### 3.5 Multi-level State Decomposition and Enhanced Reading

**Proposition 5** (Necessity of Multi-level State Decomposition). *Let the storage space be fixed. A* multi-level decomposition *with m levels is defined as follows:*

- *For each level $i = 1, \ldots, m$, there is a vocabulary matrix $\boldsymbol{C}^{(i)} \in \mathbb{R}^{V_i \times d}$, where $V_i$ is the number of prototype vectors at that level.*

- *Any vector $\boldsymbol{x} \in \mathbb{R}^d$ is represented by a sum of one prototype from each level:*

$$\hat{\boldsymbol{x}} = \sum_{i=1}^m \boldsymbol{C}_{k_i}^{(i)},$$

  *where $k_i \in \{1, \ldots, V_i\}$ is the index of the selected prototype from the i-th vocabulary.*

*The storage mechanism follows the recursive update:*

$$\boldsymbol{S}_t^{k,(i)} = diag\left(1 - \bar{f}^{(i)}(\boldsymbol{k}_t^{(i)})^\top\right)\boldsymbol{S}_{t-1}^{k,(i)} + \bar{f}^{(i)}(\boldsymbol{k}_t^{(i)})^\top \boldsymbol{k}_t^{(i)},$$
$$\boldsymbol{k}_t^{(i+1)} = \boldsymbol{k}_t^{(i)} - \bar{f}^{(i)}(\boldsymbol{k}_t^{(i)})\boldsymbol{S}_t^{k,(i)},$$

*where $\bar{f}^{(i)}$ is a composite function that queries each level of the vocabulary or state using the k vector currently to be stored and applies a normalized nonlinear transformation; the nonlinear function is typically the softmax or sigmoid function, among others. For example: $\bar{f}^{(i)}(\boldsymbol{k}_t^{(i)}) = \sigma(\boldsymbol{k}_t^{(i)}\boldsymbol{C}^{(i)\top})$ and $\boldsymbol{C}^{(i)} = \boldsymbol{S}_t^{k,(i)}$.*

*Using the iterative method described above, we can draw the following conclusion:*

1. **Representation capacity:** *The maximum number of distinct vectors that can be represented is $\prod_{i=1}^m V_i$. If all levels have the same vocabulary size $V$, this equals $V^m$.*

2. **Error bound:** *For any vector $\boldsymbol{x}$ and its optimal multi-level representation $\hat{\boldsymbol{x}}$ (with prototypes chosen greedily or jointly), the mean squared approximation error satisfies*

$$\mathbb{E}\big[\|\boldsymbol{x} - \hat{\boldsymbol{x}}\|_2^2\big] \leq \|\boldsymbol{x}\|_2^2 \cdot \prod_{i=1}^m \epsilon_i,$$

  *where $\epsilon_i$ is the expected squared error of quantizing the residual at level $i$ using $\boldsymbol{C}^{(i)}$.*

*A key trade-off exists: increasing the number of levels reduces storage error (exponentially, as $\prod \epsilon_i$) but decreases computational efficiency due to the serial computation required between levels.*

We emphasize that previous approaches employ overly simplistic reading mechanisms, typically using direct matrix multiplication between queries $q$ and states. This simplicity constitutes a significant factor (besides storage limitations) contributing to the performance gap with Softmax Attention. Our work is the first to clearly identify enhanced reading mechanisms as crucial for improving linear attention and bridging this performance gap. This mechanism implements a hierarchical access pattern through multiple channels; by comparison, the GSA reading mechanism $\text{Softmax}(q_t S^k{}_t) S^v{}_t$ represents the simplest form of indirect reading. Our enhanced version replaces the Softmax with a sigmoid activation followed by learned transformations: $(\sigma(q_t S^k{}_t) W_\sigma) S^v{}_t$, where $\sigma(x) = \frac{1}{1+e^{-x}}$. Further extending to multiple reading channels: $(q_t W_r + \sigma(q_t S^k{}_t) W_\sigma) S^v{}_t$. This approach, which is equivalent to MVA's first-order vocabulary case, demonstrates progressive performance improvement (Table 4). With multi-level vocabularies, multiple vocabularies interactions show even greater improvements over single-state approaches, underscoring the importance of balanced enhancement in both storage and reading capabilities.

### 3.6 Final Linear Model Update Rules

Integrating all five theoretical principles, we present the complete update formulation of the proposed linear model.

**1. Initialization**

**Initial conditions:**

$$q_t = f_p(x_t W_Q, r_t^{(i)}), k_{pt} = f_p(x_t W_K, r_t^{(i)}), k_t^{(0)} = x_t W_K \in \mathbb{R}^{1 \times d}, v_t^{(0)} = x_t W_V \in \mathbb{R}^{1 \times d},$$

**Initial states:**

$$S^{k^{(i)}}_0 = 0 \in \mathbb{R}^{m \times d}, n_0^{(i)} = 0 \in \mathbb{R}^{1 \times m}, E_t^{(0)} = I_m, S^{k^{(i)}}_t = S^{kv^{(i)}}_t[..., : d_k], S^{v^{(i)}}_t = S^{kv^{(i)}}_t[..., d_v :],$$

**2. Memory Assignment and Normalization.**

Here, $f^{(i)}(k_t^{(i)})$ denotes the retrieval score of the current input with respect to stored states, and $n_t^{(i)}$ represents the effective accumulation count. During storage, we normalize the contribution of the current input against historical states using $\bar{f}^{(i)}(k_t^{(i)})$. This storage mechanism is primarily derived from Theory 1, while the state size control is governed by Theories 4 and 5.

$$f^{(i)}(k_t^{(i)}) = \sigma\left(S^{k^{(i)}}_{t-1} k_t^{(i)\top}\right)^\top, \quad n_t^{(i)} = n_{t-1}^{(i)} + f^{(i)}(k_t^{(i)}), \quad \bar{f}^{(i)}(k_t^{(i)}) = \frac{f^{(i)}(k_t^{(i)})}{\max(n_t^{(i)}, 1)}. \tag{11}$$

$$m_t^{(i)} = \{k_t^{(i)}, v_t^{(i)}\}_{\dim=-1}, S^{kv^{(i)}}_t = \text{diag}\left(1 - \bar{f}^{(i)}(k_t^{(i)})^\top\right) S^{kv^{(i)}}_{t-1} + \bar{f}^{(i)}(k_t^{(i)})^\top m_t^{(i)}. \tag{12}$$

**3. Positional State Update and Residual Propagation (Theories 2 & 3).**

The positional state $S^{p^{(i)}}_t$ decouples and compresses positional information, which is subsequently integrated with the position-free KV state into a unified compressed representation.

$$S^{p^{(i)}}_t = S^{p^{(i)}}_{t-1} + f^{(i)}(k_t^{(i)})^\top (k_{pt} - k_t^{(0)}). \tag{13}$$

**4. Cross-Level Coupling and Attention Statistics.**

Using the upper bound derived in Theory 4, we determine the number of hierarchical levels required by Theory 5 as well as the state size at each level. Hierarchical decomposition is performed by computing the information loss between the current-level input and its projection onto the fixed-capacity stored state; this residual serves as the input of the next level $m_t^{(i+1)}$. After hierarchical partitioning, we compute the inter-level connection state $R_t^{(i+1)}$. During retrieval, correct memory routing is ensured via $a_t^{(i)}$. Multi-channel

enhancements can be incorporated, e.g., $q_t W_r$ and $\sigma(q_t S^{k(i)}_t)W_\sigma$. Finally, $c^{(i)}_t$ corresponds to the Taylor-approximation-based information compression component derived in Theory 3, which can be interpreted as an equivalent form of count and linear positional information, and may be replaced by more advanced data-dependent dynamic decay mechanisms.

$$m^{(i+1)}_t = m^{(i)}_t - f^{(i)}(k^{(i)}_t)S^{kv(i)}_t. \tag{14}$$

$$e^{(i)}_t = q_t W_r + \sigma(q_t S^{k(i)}_t)W_\sigma, \quad R^{(i+1)}_t\Big[f(k^{(i)}_t), f^{(i+1)}(k^{(i)}_t)\Big] = 1, \quad a^{(i)}_t = e^{(i)}_t R^{(i)\top}_t, \quad c^{(i)}_t = n^{(i)}_t + q_t S^{p(i)\top}_t. \tag{15}$$

**5. Final Aggregation and Output (Theories 4 & 5).**

The final output is obtained by normalizing the hierarchical attention statistics $b^{(i)}_t$ and aggregating them through the inter-level topological connections.

$$b^{(i)}_t = \frac{\exp\Big(\sum_i \ln(a^{(i)}_t)\Big)}{a^{(i)}_t} + e^{(i)}_t, T^{(i)}_t = R^{(i)}_t\left(S^{v(i)}_t \cdot e^{(i)\top}_t \cdot c^{(i)\top}_t\right), o_t = \sum_i \frac{b^{(i)}_t}{b^{(i)}_t \cdot e^{(i)}_t \cdot c^{(i)}_t} T^{(i)}_t. \tag{16}$$

# 4 Experiments

In this paper, we explore experiments related to converting LLMs to linear models through weight inheritance, providing experimental support for each of the five theoretical principles presented in our methodology. In the final section, we integrate these principles into Path-optimized Linear Attention (PLA) and demonstrate the effectiveness of our approach through comprehensive experiments.

We use the lm-evaluation-harness Gao et al. (2024) tool and the LongBench dataset for evaluation. For fine-tuning, we utilize LoRA Hu et al. (2021) to achieve efficient parameter updates, significantly reducing computational resources. Detailed configurations are specified in each subsection. For baseline comparisons, we compare against state-of-the-art methods including MVA and GSA, as well as GLA, RetNet, and SUPRA Mercat et al. (2024), which were benchmarked in the GSA paper.

## 4.1 Experimental Validation of Theoretical Principles

We first conduct experiments with different parameters for each theoretical principle. The evaluation uses the passkey retrieval task, standard benchmarks from lm-evaluation-harness (ARC-C and MMLU), and the long-sequence SAMSUM dataset from LongBench for testing and guidance.

**Theory 1: Redundancy Elimination with Similarity Thresholds.** Table 1 shows experiments with

Table 1: Results for Theory 1 with different similarity thresholds

| Method | Finetune Tokens | Passkey (1K-8K) | ARC | MMLU | SAMSUM |
|--------|-----------------|-----------------|-----|------|--------|
| Mistral-7B-v0.1 | – | 100.0 | 54.0 | 62.4 | 43.6 |
| Theory 1 ($t = 1$) | 200M | 100.0 | 54.0 | 62.4 | 43.6 |
| Theory 1 ($t = 0.95$) | 200M | 100.0 | 53.4 | 60.7 | 42.9 |
| Theory 1 ($t = 0.9$) | 200M | 100.0 | 51.8 | 57.6 | 41.5 |
| Theory 1 ($t = 0.8$) | 200M | 100.0 | 49.7 | 49.2 | 38.3 |

different similarity thresholds for Theory 1, where $t = 1$ indicates the threshold used for similarity discrimination in Theory 1 (e.g., $t = 0.9$ means tokens with cosine similarity $\geq 0.9$ are considered identical). Results demonstrate that when similarity exceeds a certain level (e.g., 0.95), performance approaches that of the original model. This approach functions as a quantization process, indicating model insensitivity to token variations within certain ranges.

**Theories 2 & 3: Positional Information Decoupling and Compression.**

Table 2 presents experiments for Theories 2 and 3, exploring positional information separation, compression, and refined Taylor expansion approaches. Here, "depos" indicates decoupled positional encoding, while "de&cprpos" indicates decoupled and compressed positional encoding.

Table 2: Results for Theories 2 & 3 with different positional encoding strategies

| Method | Finetune Tokens | Passkey (1K-8K) | ARC | MMLU | SAMSUM |
|---|---|---|---|---|---|
| Theory 2&3 ($t = 0.95$, w/ depos) | 500M | 100.0 | 52.8 | 58.6 | 43.5 |
| Theory 2&3 ($t = 0.95$, w/ de&cprpos) | 500M | 100.0 | 50.2 | 53.1 | 40.7 |
| Theory 2&3 ($t = 0.95$, w/ de&cprpos-2) | 500M | 100.0 | 51.9 | 55.7 | 42.7 |

**Theory 4: Inter-layer Scaling Factors.**

Table 3 shows experiments for Theory 4 with different layer-wise scaling factors. Using parameters from previous theories ($t = 0.95$, w/ depos), $c_{\text{scale}} = 1.2$ indicates that the KV cache size at layer $l + 1$ is 1.2 times that of layer $l$, up to the midpoint of the total layers, after which the KV cache size remains constant. $c_{\text{scale-l8}} = 1.2$ & $c_{\text{scale-l16}} = 1.6$ indicates a scaling factor of 1.2 for the first 8 layers and 1.6 for layers 8-16.

Table 3: Results for Theory 4 with different layer scaling factors

| Method | Finetune Tokens | Passkey (1K-8K) | ARC | MMLU | SAMSUM |
|---|---|---|---|---|---|
| Theory 4 (w/ $c_{\text{scale}} = 1.2$) | 500M | 100.0 | 46.2 | 52.0 | 40.8 |
| Theory 4 (w/ $c_{\text{scale}} = 1.4$) | 500M | 100.0 | 51.7 | 57.8 | 43.2 |
| Theory 4 (w/ $c_{\text{scale}} = 1.6$) | 500M | 100.0 | 53.4 | 60.1 | 42.7 |
| Theory 4 (w/ $c_{\text{l8}} = 1.2$ & $c_{\text{l16}} = 1.6$) | 500M | 100.0 | 53.3 | 59.7 | 42.9 |

Figure A illustrates the evolution of KV cache length across layers as predicted by Theory 4.

**Theory 5: Enhanced Reading Mechanisms and Multi-state Configurations.**

Table 4 presents experiments for Theory 5, examining various enhanced reading mechanisms and different state sizes. We use single and two-level vocabulary configurations, without positional separation for faster convergence, focusing solely on reading mechanism variations. Results show progressive performance improvement with enhanced reading capabilities, with our PLA approach building upon GSA by adding multi-channel reading and multi-state interaction, equivalent to incorporating vocabulary interaction(VI) into MVA.

Table 4: Results for Theory 5 with different reading mechanisms

| Method | Finetune Tokens | Passkey (2K) | ARC | MMLU | SAMSUM |
|---|---|---|---|---|---|
| Theory 5 (GSA) | 500M | 0.0 | 31.7 | 22.3 | 18.9 |
| Theory 5 (GSA w/ sigmoid) | 500M | 0.0 | 33.5 | 23.5 | 18.9 |
| Theory 5 (GSA + MetaLA) | 500M | 10.0 | 35.6 | 24.1 | 21.7 |
| Theory 5 (MVA) | 500M | 20.0 | 38.2 | 25.6 | 24.9 |
| Theory 5 (PLA: MVA + VI) | 500M | 40.0 | 39.4 | 26.2 | 24.7 |

## 4.2 PLA: Integrated Path-optimized Linear Attention

Building upon the experimental validation of individual theoretical principles, we now present the integrated PLA model that combines all five theoretical components into a unified framework. PLA also employs a two-level vocabulary decomposition, similar to GSA and MVA whose state is 128 in size, and we their basis by adding operations such as positional encoding decoupling, read enhancement, and lexicon interaction.

The experimental results demonstrate that PLA achieves state-of-the-art performance while maintaining competitive efficiency. The integrated approach successfully leverages all five theoretical principles to create a robust linear attention mechanism that narrows the performance gap with softmax attention.

Using the proposed method, we fine-tune pretrained model weights of varying scales and architectures, including Mistral-7B, Qwen2.5-3B, Qwen2.5-14B, and LLaMA3-70B, to obtain our PLA models.

**Fine-tuning Setup.** We adopt the LoRA method for fine-tuning. LoRA is applied to all linear projection weights in the Transformer architecture. The rank of the low-rank adaptation matrices is set to 128. Training

Table 5: Comprehensive evaluation of PLA against state-of-the-art methods

| Model | Size | Tokens | ARC-e | ARC-c | Hella. | MMLU | Avg. |
|---|---|---|---|---|---|---|---|
| *Models trained from scratch (reference)* | | | | | | | |
| RWKV6 | 7B | 1.4T | 73.6 | 44.0 | 75.2 | 43.9 | 58.0 |
| Mamba | 7B | 1.2T | 77.6 | 46.8 | 77.8 | 33.2 | 60.0 |
| Llama2 | 7B | 2T | 76.4 | 46.2 | 76.0 | 45.5 | 60.2 |
| Mistral | 7B | ? | 80.8 | 54.0 | 81.1 | 62.4 | 66.6 |
| *Models via fine-tuning* | | | | | | | |
| SUPRA | 7B | +20B | 74.6 | 42.3 | 74.8 | 28.0 | - |
| RetNet | 7B | +20B | 73.3 | 39.9 | 72.9 | 26.1 | 51.9 |
| GLA | 7B | +20B | 74.6 | 44.0 | 75.9 | 28.4 | 56.5 |
| GSA | 7B | +20B | 75.9 | 43.9 | 76.5 | 32.4 | 57.7 |
| MVA | 7B | +10B | 78.3 | 47.5 | 78.1 | 34.4 | 60.3 |
| PLA (Ours) | 7B | +8B | 78.5 | 47.2 | 78.3 | 42.1 | 61.3 |

Table 6: Zero-shot and few-shot evaluation results on common language understanding benchmarks. ARC-c/e, HellaSwag, PIQA, and WinoGrande are evaluated in zero-shot setting, while MMLU uses 5-shot. AVG denotes the average accuracy across all tasks.

| Model | ARC-c | ARC-e | HellaSwag | PIQA | WinoGrande | MMLU | AVG |
|---|---|---|---|---|---|---|---|
| RWKV7-2.9B[1] | 48.7 | 81.0 | 76.4 | 79.7 | 72.8 | 55.0 | 68.93 |
| GatedDeltaNet-1.3B[1] | 38.4 | 71.2 | 55.7 | 72.2 | 57.4 | 32.4 | 54.55 |
| Mamba-2-2.7B[1] | 36.4 | 69.6 | 66.6 | 76.4 | 64.0 | 33.6 | 57.77 |
| Qwen2.5-3B (Original) | 45.0 | 77.4 | 73.5 | 78.6 | 68.5 | 65.7 | 68.12 |
| Qwen2.5-3B-PLA (ft: 10B tokens) | 45.2 | 78.0 | 71.4 | 78.2 | 66.8 | 39.2 | 63.13 |
| Qwen2.5-14B (Original) | 58.9 | 82.3 | 82.9 | 82.3 | 75.3 | 79.7 | 76.90 |
| Qwen2.5-14B-PLA (ft: 10B tokens) | 60.2 | 83.2 | 81.7 | 82.5 | 74.2 | 61.9 | 73.95 |
| LLaMA-3-70B | 64.1 | 86.9 | 81.6 | 84.8 | 80.6 | 82.0 | 80.00 |
| LLaMA-3-70B-GSA (ft: 12.5B tokens) | 47.6 | 72.6 | 74.9 | 64.5 | 59.4 | 42.1 | 60.18 |
| LLaMA-3-70B-PLA (ft: 10B tokens) | 65.2 | 87.8 | 79.8 | 85.6 | 78.9 | 58.4 | 75.95 |

Table 7: Experimental results on long-context benchmarks, training efficiency comparison and passkey task

| Model | Qasper | NarrativeQA | QMSum |
|---|---|---|---|
| *Models trained from scratch* | | | |
| RWKV6 | 9.2 | 14.4 | 1.1 |
| Mamba | 5.6 | 27.9 | 0.8 |
| Mistral | 25.8 | 25.1 | 5.0 |
| *Fine-tuned from Mistral-7B (10B tokens)* | | | |
| RetNet | 11.1 | 0.0 | 0.0 |
| GLA | 18.4 | 17.2 | 9.0 |
| GSA | 18.8 | 19.2 | 10.0 |
| MVA | 20.7 | 20.4 | 9.58 |
| PLA | 22.3 | 21.2 | 10.7 |

| Method | Memory / Time |
|---|---|
| MetaLA | 36,317 MiB / 75.08 s/it |
| GSA | 37,619 MiB / 81.67 s/it |
| MVA w/ VD | 40,096 MiB / 105.79 s/it |
| PLA w/ VD | 41,278 MiB / 118.67 s/it |

| Model/passkey task | 256 | 512 | 1024 | 2048 | 4096 | 8192 |
|---|---|---|---|---|---|---|
| Qwen | 1.0 | 1.0 | 1.0 | 1.0 | 1.0 | 1.0 |
| GSA | 1.0 | 0.8 | 0.7 | 0.5 | 0.3 | 0.4 |
| **PLA** | **1.0** | **1.0** | **1.0** | **1.0** | **1.0** | **0.9** |

Table 8: Performance on LongBench tasks.

| Model | SQA | MQA | Sum | FS | Code | Avg. |
|---|---|---|---|---|---|---|
| Qwen2.5-7B | 28.60 | 28.71 | 17.85 | 69.51 | 58.60 | 40.65 |
| PLA-7B-Qwen (Ours) | 24.89 | 23.56 | 18.92 | 68.07 | 58.59 | 38.80 |
| Qwen2.5-14B | 36.25 | 33.85 | 20.12 | 74.60 | 63.42 | 45.65 |
| PLA-14B-Qwen (Ours) | 34.32 | 31.22 | 17.02 | 73.96 | 63.58 | 44.02 |

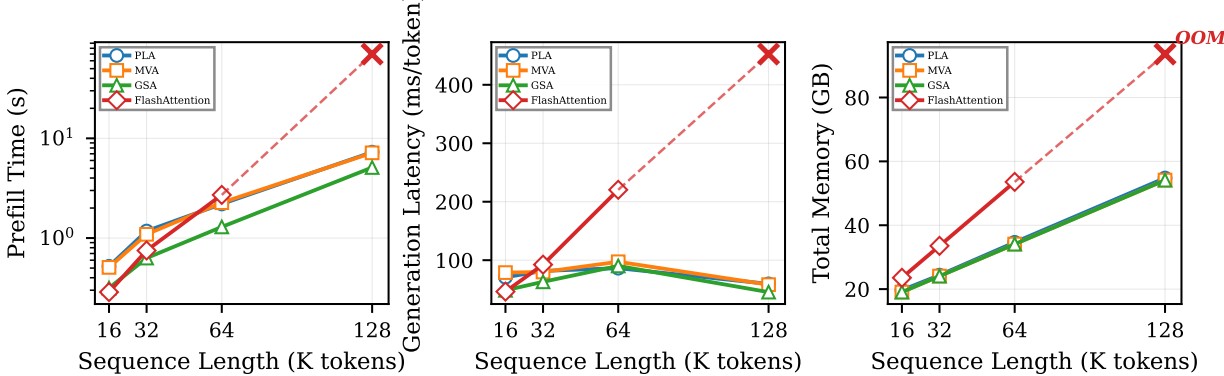

Figure 2: Inference efficiency comparison of PLA, MVA, GSA, and FlashAttention under different sequence lengths. OOM indicates out-of-memory errors.

uses a constant learning rate of $1 \times 10^{-4}$ with a warmup of 100 steps. Each training step processes a batch size of 0.5M tokens, and the total number of fine-tuning tokens is 10B.

**Common-Sense Reasoning**

Table 5 reports the performance of PLA based on Mistral weights on common-sense reasoning benchmarks. Our method achieves superior performance compared to GLA, GSA, and MVA.

To further demonstrate the scalability and generality of PLA, we fine-tune Qwen2.5-3B, Qwen2.5-14B, and LLaMA3-70B models. With only 10B fine-tuning tokens, PLA is able to recover nearly 95% of the original Transformer performance on common-sense reasoning tasks, as shown in Table 6.

**Long-Sequence Tasks**

Tables 7 and 8 present the performance of PLA models with different parameter scales on LongBench and various long-sequence tasks.

Compared to the original Attention-based models without length extension, PLA surpasses the original models on certain long-sequence benchmarks, demonstrating its ability to improve long-context modeling capacity.

**Retrieval Tasks**

Table 7 also reports performance on the Passkey retrieval task. With 8K-length fine-tuning, PLA is able to almost perfectly solve the 8K-length retrieval task, a capability that previous linear attention models struggled to achieve.

**Efficiency Analysis**

Figure 4.2 compares the inference speed of two-level PLA with Transformer and GSA. PLA achieves a speed close to GSA under the same state size, while significantly outperforming GSA in accuracy.

Table 9: Performance comparison of different models across various benchmarks.

| Model | Training Tokens (B) | PiQA | ARC-e | ARC-c (norm) | HellaSwag (norm) | Winogrande | MMLU (5-shot) | Avg. | Avg. (no MMLU) |
|---|---|---|---|---|---|---|---|---|---|
| Mistral 7B | – | 82.1 | 80.9 | 53.8 | 81.0 | 74.0 | 62.4 | 72.4 | 74.4 |
| Mistral 7B SUPRA | 100 | 80.4 | 75.9 | 45.8 | 77.1 | 70.3 | 34.2 | 64.0 | 69.9 |
| Mistral 7B LoLCATs Zhang et al. (2025) | **0.04** | 81.5 | **81.7** | 54.9 | 80.7 | 74.0 | 51.4 | 70.7 | 74.5 |
| Mistral 7B PLA-SWA (Ours) | **0.04** | **82.1** | 81.5 | **55.4** | **81.1** | **74.2** | **54.2** | **71.4** | **74.9** |
| Llama 3 8B | – | 79.9 | 80.1 | 53.3 | 79.1 | 73.1 | 66.6 | 72.0 | 73.1 |
| Llama 3 8B Hedgehog | 0.04 | 77.4 | 71.1 | 40.6 | 66.5 | 54.3 | 24.2 | 55.7 | 62.0 |
| Llama 3 8B LoLCATs | 0.04 | 80.9 | 81.7 | **54.9** | 79.7 | **74.1** | 52.8 | 70.7 | 74.2 |
| Llama 3 8B PLA-SWA (Ours) | 0.04 | 81.2 | **82.8** | 54.5 | 79.4 | 74.1 | 53.8 | **71.0** | **74.4** |

Table 10: Performance comparison of different attention variants across benchmarks.

| Model | Training Tokens (B) | Params | PIQA | Hella. | Wino. | ARC-e | ARC-c | Avg. |
|---|---|---|---|---|---|---|---|---|
| std att | 50 | 480M | $66.4 \pm 0.1$ | $40.5 \pm 0.1$ | $52.3 \pm 0.7$ | $52.8 \pm 0.4$ | $29.1 \pm 0.5$ | 48.21 |
| sw-nope | 50 | 480M | $66.7 \pm 0.4$ | $40.9 \pm 0.1$ | $52.6 \pm 0.6$ | $53.1 \pm 0.1$ | $28.4 \pm 0.4$ | 48.35 |
| sw-ovq Alonso et al. (2026) | 50 | 480M | $66.6 \pm 0.2$ | $41.1 \pm 0.2$ | $52.4 \pm 0.2$ | $52.7 \pm 0.1$ | $28.7 \pm 0.4$ | 48.30 |
| PLA-380M (Ours) | **25** | **380M** | **67.6** | 40.1 | **52.6** | **57.2** | 28.6 | **49.22** |

## 5 Conclusion

We chart the optimal path from Softmax to linear attention and verify, both theoretically and empirically, the pivotal roles of (i) redundancy removal, (ii) positional-code disentanglement & compression, (iii) tokenizer vocabulary reuse, (iv) layer-wise similarity, and (v) multi-vocabulary decomposition. Leveraging these insights, PLA sets a new efficiency-performance frontier: it matches or surpasses the best existing linearized models while consuming equal or fewer training tokens, offering a ready-to-use recipe for compressing large-language-model attention into a fixed-size, linear-complexity operator.

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

# A   Appendix

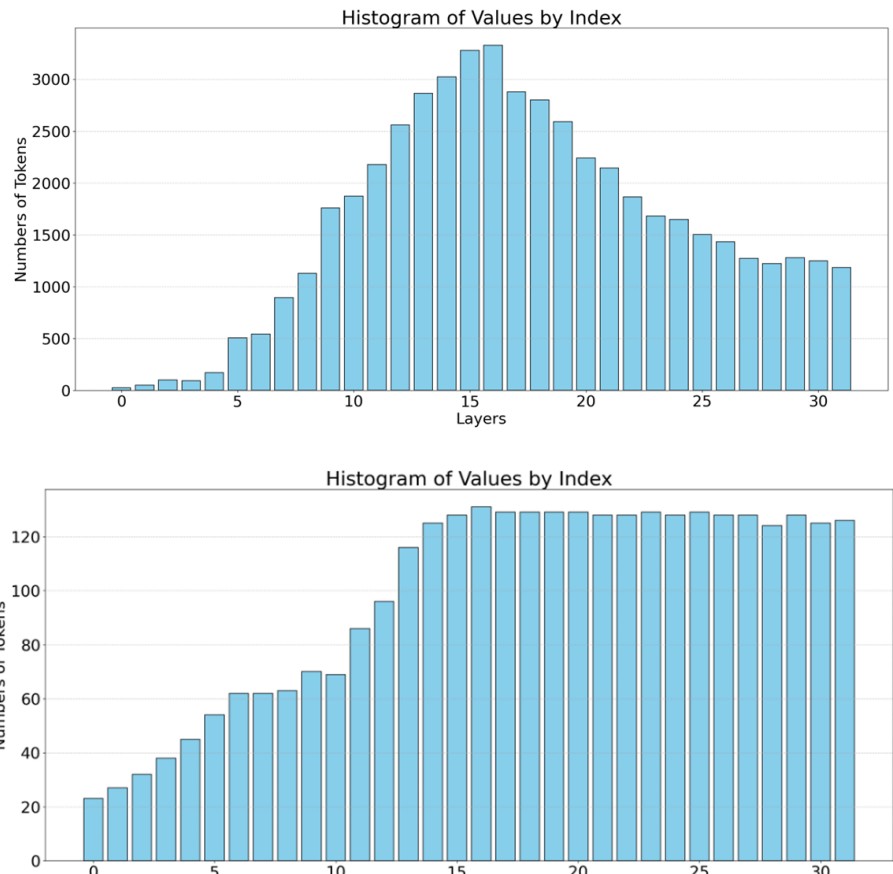

Figure 3:    The task uses passkey task.  The above figure shows the result of KV cache size per layer obtained by applying Theory 1 for compression and then the following figure shows the result obtained by adding Theory 4 for interlayer similarity, after interlayer similarity the KV cache of the subsequent layers is controlled.

## A.1   Proof of Proposition 1

### A.1.1   Proof of the Storage Upper Bound

Without loss of generality, we first consider the case where both batch size and number of heads equal 1. Assume that the numeric type has bit-width $b$ and the head dimension is $d$.  For the infinitely growing key and value sequences, denote them as matrices of shape $\mathbb{R}^{t \times d}$, and concatenate them along the feature dimension into

$$\boldsymbol{S}^{KV} \in \mathbb{R}^{t \times (d_k + d_v)}, \quad \boldsymbol{S}^K = \boldsymbol{S}^{KV}[:, : d_k], \quad \boldsymbol{S}^V = \boldsymbol{S}^{KV}[:, d_k : d_k + d_v].$$

Let the count matrix be $\boldsymbol{c} \in \mathbb{R}^{t \times 1}$.

*Proof.* Since each numerical component is represented with $b$ bits, each component can take $2^b$ distinct values.  The dimensions of a vector $\mathbf{u} \in \mathbb{R}^{d_k + d_v}$ are independent; therefore, the total number of possible distinct vectors $C$ is

$$\prod_{m=1}^{d_k + d_v} 2^b = 2^{b(d_k + d_v)}.$$

Table 11: Summary of Theoretical Principles for Softmax-to-Linear Attention Compression

| Theory | Mechanism | Function | Error Bound | Equivalent Mechanism |
|--------|-----------|----------|-------------|----------------------|
| **T1** | Redundancy Elimination/Quantization | Controls state upper bound | Lossless | Delta Rule, Quantization |
| **T2** | Positional Information Decoupling | Reduces first-layer state count | Lossless | Tokenizer vocabulary + positional decoupling |
| **T3** | Positional Information Compression | Controls positional dimension error | $O\left(\frac{n}{\text{base}^{2j/d}}\right)$ | Gating mechanism |
| **T4** | Inter-layer Similarity | Controls intermediate layer state count | Controllable (empirically validated) | Residual connection + similarity propagation |
| **T5** | Multi-level State Decomposition | Fixed-size state representation | Exponential decay | Vocabulary decomposition + enhanced reading |

The distinct key-value pairs appearing in the sequence must belong to a subset of this finite set, hence $C \leq 2^{b(d_k+d_v)}$. Storing each unique vector requires $b(d_k + d_v)$ bits, and each count requires $b_c$ bits. Thus, the upper bound on total storage $S_b$ is $C \times (b(d_k + d_v) + b_c)$, and combining this with the bound on $C$ yields the desired inequality. $\square$

### A.1.2 Proof of Attention Equivalence

*Proof.* Let $\mathcal{I}_i = \{t \mid (\mathbf{k}_t, \mathbf{v}_t) = (\mathbf{k}_i, \mathbf{v}_i)\}$ be the set of time indices where the $i$-th unique key-value pair appears; then $|\mathcal{I}_i| = c_i$. The original attention output can be written as

$$\mathbf{o} = \sum_{t=1}^{T} \frac{e^{\mathbf{q}^\top \mathbf{k}_t}}{Z}\mathbf{v}_t, \quad Z = \sum_{t=1}^{T} e^{\mathbf{q}^\top \mathbf{k}_t}.$$

Grouping the summation by unique pairs:

$$\mathbf{o} = \sum_{i=1}^{C} \sum_{t \in \mathcal{I}_i} \frac{e^{\mathbf{q}^\top \mathbf{k}_t}}{Z}\mathbf{v}_t.$$

Since for any $t \in \mathcal{I}_i$, we have $\mathbf{k}_t = \mathbf{k}_i$ and $\mathbf{v}_t = \mathbf{v}_i$, it follows that

$$\sum_{t \in \mathcal{I}_i} \frac{e^{\mathbf{q}^\top \mathbf{k}_t}}{Z}\mathbf{v}_t = c_i \frac{e^{\mathbf{q}^\top \mathbf{k}_i}}{Z}\mathbf{v}_i.$$

Moreover, the denominator can be expressed as $Z = \sum_{i=1}^{C} \sum_{t \in \mathcal{I}_i} e^{\mathbf{q}^\top \mathbf{k}_t} = \sum_{i=1}^{C} c_i e^{\mathbf{q}^\top \mathbf{k}_i}$. Substituting these expressions yields

$$\mathbf{o} = \sum_{i=1}^{C} c_i \frac{e^{\mathbf{q}^\top \mathbf{k}_i}}{\sum_{j=1}^{C} c_j e^{\mathbf{q}^\top \mathbf{k}_j}}\mathbf{v}_i,$$

which is precisely the attention output computed from the compressed representation. This completes the proof. $\square$

### A.1.3 Comparison with the Delta Rule Formulation

*Proof.* We now describe the KV cache storage process using the deduplication mechanism in terms of the update rules of linear attention. Consider the unique-filtered sequence as the state $\boldsymbol{S}_t^{(K)}$, where $t$ denotes the current time step or the row index up to which the sequence has been processed. $\boldsymbol{S}_t^{(K)}$ represents the KV cache maintained at time $t$. For linear attention, the size of $\boldsymbol{S}_t^{(K)}$ is fixed; for standard KV cache, each new element is concatenated to the previous cache. For the deduplication-based KV cache, its update rule can be expressed as:

$$\Delta(\boldsymbol{k}_t) = \boldsymbol{k}_t - Q\big(\boldsymbol{k}_t \boldsymbol{S}_{t-1}^{(K)\top}\big)\,\boldsymbol{S}_{t-1}^{(K)}, \qquad \Delta(\boldsymbol{v}_t) = \boldsymbol{v}_t - Q\big(\boldsymbol{v}_t \boldsymbol{S}_{t-1}^{(V)\top}\big)\,\boldsymbol{S}_{t-1}^{(V)}.$$

If $\Delta(\boldsymbol{k}_t) \le 1 - \text{threshold}$, then $\boldsymbol{S}_t^{(K)} = \boldsymbol{S}_{t-1}^{(K)}$. Otherwise, if $\Delta(\boldsymbol{k}_t) > 1 - \text{threshold}$, we update via concatenation:

$$\boldsymbol{S}_t^{(K)} = \text{concat}\big(\boldsymbol{S}_{t-1}^{(K)}, \boldsymbol{k}_t\big).$$

For the unique operation, the effective threshold is 1. And the $Q$ function is defined as:

$$Q(a_{ij}) = \begin{cases} 1 & \text{if } a_{ij} \text{ is max} \\ 0 & \text{otherwise} \end{cases} \tag{17}$$

When the state size is manually limited to $m$, as in linear attention, once $\boldsymbol{S}_t^{(K)}$ reaches size $m$, further growth is prohibited. In this case, information differences must be integrated into the previous state via gating, yielding an update analogous to the Delta Rule:

$$\boldsymbol{S}_t^{(K)} = \big(1 - \beta \cdot Q(\boldsymbol{k}_t \boldsymbol{S}_{t-1}^{(K)\top})^\top\big)\,\boldsymbol{S}_{t-1}^{(K)} - \beta \cdot Q(\boldsymbol{k}_t \boldsymbol{S}_{t-1}^{(K)\top})^\top \Delta(\boldsymbol{k}_t).$$

When the key–value pairs are stored jointly as matrix states, the update becomes

$$\boldsymbol{S}_t^{(KV)} = \big(1 - \beta \cdot Q(\boldsymbol{k}_t \boldsymbol{S}_{t-1}^{(K)\top})^\top\big)\,\boldsymbol{S}_{t-1}^{(KV)} - \beta \cdot Q(\boldsymbol{k}_t \boldsymbol{S}_{t-1}^{(K)\top})^\top \Delta(\boldsymbol{v}_t),$$

which is essentially equivalent to the Delta Rule update from the fast-weight literature, except that $\phi(\boldsymbol{k}_t)$ is replaced by $Q(\boldsymbol{k}_t \boldsymbol{S}_{t-1}^{(K)})$. Moreover, MVA demonstrates that $Q(\boldsymbol{k}_t \boldsymbol{S}_{t-1}^{(K)})$ can be substituted with $Q(\boldsymbol{k}_t \boldsymbol{W}_c)$ to achieve comparable performance, and with carefully chosen $Q$ functions, this becomes exactly equivalent to using $\phi(\boldsymbol{k}_t)$. $\qquad\square$

### A.2 Proof of Proposition 2

We provide a constructive proof.

### A.2.1 Construction of the compressed representation

*Proof.* Given the original first-layer input $\tilde{\boldsymbol{X}}^{(1)}$, we construct the triple $(\boldsymbol{E}, \boldsymbol{I}, \tilde{\boldsymbol{P}})$ as follows:

- **Vocabulary embedding matrix $\boldsymbol{E}$**: Directly take the pre-trained word embedding matrix $\boldsymbol{E}_{\text{vocab}}$. Since this matrix remains unchanged during inference, it does not need to be stored repeatedly in the compressed representation (it can be considered as shared parameters).

- **Index vector $\boldsymbol{I}$**: For each position $t$, record its token index $w_t$, i.e. $\boldsymbol{I}_t = w_t$. Because $w_t \in \{1, \ldots, V\}$, each index can be stored using $\lceil \log_2 V \rceil$ bits.

- **Positional information matrix $\tilde{\boldsymbol{P}}$**: To reconstruct $\tilde{\boldsymbol{X}}^{(1)}$ from $(\boldsymbol{E}, \boldsymbol{I})$, we need to retain the effect of positional encoding. Define

$$\tilde{\boldsymbol{P}} = \text{PE}^{-1}\big(\tilde{\boldsymbol{X}}^{(1)}, \; \boldsymbol{E}[\boldsymbol{I}, :]\big),$$

where $\text{PE}^{-1}$ is the inverse operation of the positional encoding function (i.e., given the output and the word embeddings, recover the positional information). For common positional encoding schemes, $\text{PE}^{-1}$ exists and is easy to compute:

- For absolute additive encoding: $\tilde{\boldsymbol{X}}^{(1)} = \boldsymbol{E}[\boldsymbol{I}, :] + \boldsymbol{P}$, then $\tilde{\boldsymbol{P}} = \tilde{\boldsymbol{X}}^{(1)} - \boldsymbol{E}[\boldsymbol{I}, :] = \boldsymbol{P}$.
- For RoPE, we do not need to reconstruct the original sequence explicitly; instead we reconstruct the computation equivalent to attention losslessly, which yields a practically usable compression ratio. For completeness, we also sketch the reconstruction of the original sequence.

$\square$

### A.2.2 Lossless reconstruction (RoPE case)

*Proof.* Consider an input sequence of length $T$ with token indices $w_1, \ldots, w_T \in \{1, \ldots, V\}$. Suppose the model employs RoPE positional encoding, whose core operation is to rotate each pair of dimensions $(2j, 2j+1)$ by an angle $\theta_j = \text{base}^{-2j/d}$, $j = 0, \ldots, d/2-1$. Denote the rotation parameters as $\tilde{\boldsymbol{P}} = \Theta = \{\theta_j\}_{j=0}^{d/2-1}$. RoPE combines an arbitrary vector $\boldsymbol{x} \in \mathbb{R}^d$ with a position index $m$ into a rotated vector $\tilde{\boldsymbol{x}}_m = f_{\text{RoPE}}(\boldsymbol{x}, m, \Theta)$ defined by

$$\begin{cases} \tilde{x}_{m,2j} = x_{2j} \cos(m\theta_j) - x_{2j+1} \sin(m\theta_j), \\ \tilde{x}_{m,2j+1} = x_{2j} \sin(m\theta_j) + x_{2j+1} \cos(m\theta_j), \end{cases} \qquad j = 0, \ldots, \frac{d}{2} - 1.$$

The first-layer input is the matrix obtained by applying RoPE to the word embedding sequence:

$$\tilde{\boldsymbol{X}}^{(1)} = [\tilde{\boldsymbol{e}}_{w_1,1}, \ldots, \tilde{\boldsymbol{e}}_{w_T,T}]^\top \in \mathbb{R}^{T \times d},$$

where $\tilde{\boldsymbol{e}}_{w_t,t} = f_{\text{RoPE}}(\boldsymbol{e}_{w_t}, t, \Theta)$.

To verify that the compressed representation $(\boldsymbol{E}, \boldsymbol{I}, \Theta)$ can losslessly recover $\tilde{\boldsymbol{X}}^{(1)}$, it suffices to note that for any position $t$, the vector $\tilde{\boldsymbol{e}}_{w_t,t}$ is uniquely determined by $\boldsymbol{e}_{w_t}$, $t$, and $\Theta$ via the RoPE definition. Hence, given $\boldsymbol{E}$, $\boldsymbol{I}$, and $\Theta$, we can reconstruct the entire $\tilde{\boldsymbol{X}}^{(1)}$ position by position, exactly matching the original definition. Thus the reconstruction is lossless.

Moreover, in subsequent attention computations we do not need to reconstruct the full matrix explicitly. For any query position $n$ and key position $s$, the (unnormalized) attention score is

$$A_{n,s} = \exp\left( \frac{1}{\sqrt{d}} \tilde{\boldsymbol{e}}_{w_n,n}^\top \tilde{\boldsymbol{e}}_{w_s,s} \right).$$

The inner product $\tilde{\boldsymbol{e}}_{w_n,n}^\top \tilde{\boldsymbol{e}}_{w_s,s}$ can be expressed using only $\boldsymbol{e}_{w_n}$, $\boldsymbol{e}_{w_s}$, the relative position $\Delta = n - s$, and the parameters $\Theta$. Indeed, from the inner product property of RoPE:

$$\tilde{\boldsymbol{e}}_{w_n,n}^\top \tilde{\boldsymbol{e}}_{w_s,s} = \sum_{j=0}^{d/2-1} \left[ (e_{w_n,2j} e_{w_s,2j} + e_{w_n,2j+1} e_{w_s,2j+1}) \cos(\Delta\theta_j) + (e_{w_n,2j} e_{w_s,2j+1} - e_{w_n,2j+1} e_{w_s,2j}) \sin(\Delta\theta_j) \right].$$

Thus, knowing the word embeddings $\boldsymbol{e}_{w_n}$, $\boldsymbol{e}_{w_s}$ (obtained from $\boldsymbol{E}$ via the indices) and the relative position $\Delta$ (computed from the position indices) together with $\theta_j$ from $\Theta$, we can compute this inner product. This means the entire attention computation can be performed directly on the compressed representation, without storing the rotated vectors for each position or reconstructing the full matrix. $\square$

### A.2.3 Upper bound on the number of distinct row vectors

*Proof.* Each row $\tilde{\boldsymbol{e}}_{w_t,t}$ of $\tilde{\boldsymbol{X}}^{(1)}$ is determined by the token index $w_t$ and the position $t$. However, $w_t$ can only take $V$ possible values. Therefore the set $\mathcal{U}$ of all rows necessarily satisfies $|\mathcal{U}| \leq V$. Indeed, for a fixed token index $v$, the vectors $\tilde{\boldsymbol{e}}_{v,t}$ produced at different positions $t$ may be different, but vectors corresponding to different tokens are always distinct (because RoPE is a linear transformation and the word embeddings are unique). Hence the total number does not exceed the vocabulary size. This bound is tight: equality holds when every token appears and each token appears at exactly one position. $\square$

### A.2.4 Storage size analysis

*Proof.*
- The vocabulary embedding matrix $\boldsymbol{E}$ occupies $V \cdot d \cdot b$ bits, but these parameters already exist in the model and need no extra storage in the compressed representation (they are merely referenced).

- The index vector $\boldsymbol{I}$ requires storing $T$ integers, each in the range $[1, V]$, hence $T \lceil \log_2 V \rceil$ bits.

- The rotation parameters $\Theta$ contain $d/2$ scalars, each of $b$ bits, totalling $(d/2) \cdot b$ bits, which is independent of the sequence length $T$.

Therefore, the total storage overhead of the compressed representation is $T \lceil \log_2 V \rceil + (d/2) \cdot b$ bits, compared to the original input which uses $T \cdot d \cdot b$ bits. This achieves a compression from $O(Td)$ to $O(T \log V + d)$, which is particularly significant when $d \gg \log V$. $\qquad\square$

### A.3 Proof of Proposition 3

### A.3.1 Error estimate of Proposition 3

*Proof.* We provide a step-by-step derivation that combines the original reasoning with necessary definitions and rigorous estimates.

**1. Notation and basic identities.** Let $\tilde{\boldsymbol{q}}_n = f_p(\boldsymbol{q}_n, \boldsymbol{p}(n))$ and $\tilde{\boldsymbol{k}}_s = f_p(\boldsymbol{k}_s, \boldsymbol{p}(s))$ denote the vectors after applying RoPE. The exact unnormalized attention score is

$$A_{n,s} = \exp\left(\frac{1}{\sqrt{d}} \tilde{\boldsymbol{q}}_n^\top \tilde{\boldsymbol{k}}_s\right).$$

Using the RoPE inner product formula,

$$\tilde{\boldsymbol{q}}_n^\top \tilde{\boldsymbol{k}}_s = \sum_{j=0}^{d/2-1} \Big[ (q_{n,2j} k_{s,2j} + q_{n,2j+1} k_{s,2j+1}) \cos(\Delta \theta_j) + (q_{n,2j} k_{s,2j+1} - q_{n,2j+1} k_{s,2j}) \sin(\Delta \theta_j) \Big],$$

where $\Delta = n - s$. For convenience we also denote the unrotated query–key inner product as

$$\tilde{\boldsymbol{q}}_n^\top \boldsymbol{k}_s = \sum_{j=0}^{d/2-1} \Big[ q_{n,2j} k_{s,2j} \cos(n\theta_j) - q_{n,2j+1} k_{s,2j} \sin(n\theta_j) + q_{n,2j} k_{s,2j+1} \sin(n\theta_j) + q_{n,2j+1} k_{s,2j+1} \cos(n\theta_j) \Big].$$

**2. Small-angle expansion.** Because base is large, $\theta_j$ is small for $j \geq 2$ under the assumption $|\Delta| \theta_j \leq 1$. We expand cos and sin:

$$\cos(\Delta \theta_j) = 1 - \frac{(\Delta \theta_j)^2}{2} + O((\Delta \theta_j)^4), \qquad \sin(\Delta \theta_j) = \Delta \theta_j + O((\Delta \theta_j)^3).$$

Substituting these into $\tilde{\boldsymbol{q}}_n^\top \tilde{\boldsymbol{k}}_s$ yields

$$\tilde{\boldsymbol{q}}_n^\top \tilde{\boldsymbol{k}}_s = \tilde{\boldsymbol{q}}_n^\top \boldsymbol{k}_s + \tilde{\boldsymbol{q}}_n^\top (\tilde{\boldsymbol{k}}_s - \boldsymbol{k}_s),$$

where $\tilde{\boldsymbol{k}}_s - \boldsymbol{k}_s$ collects the first-order effects of rotation. More explicitly,

$$\tilde{\boldsymbol{q}}_n^\top (\tilde{\boldsymbol{k}}_s - \boldsymbol{k}_s) = \sum_j \tilde{\boldsymbol{q}}_{nj}^\top \boldsymbol{k}_{sj} \Big[ (\cos(\Delta \theta_j) - 1) + \sin(\Delta \theta_j) \Big],$$

with each term behaving as $O(\Delta \theta_j)$ plus higher-order corrections.

**3. Exponential expansion.** Write the exact score as

$$A_{n,s} = \exp\left(\frac{1}{\sqrt{d}}\tilde{\boldsymbol{q}}_n^\top \boldsymbol{k}_s\right)\exp\left(\frac{1}{\sqrt{d}}\tilde{\boldsymbol{q}}_n^\top(\tilde{\boldsymbol{k}}_s - \boldsymbol{k}_s)\right).$$

Expand the second exponential using the Taylor series $e^u = 1 + u + O(u^2)$:

$$\exp\left(\frac{1}{\sqrt{d}}\tilde{\boldsymbol{q}}_n^\top(\tilde{\boldsymbol{k}}_s - \boldsymbol{k}_s)\right) = 1 + \frac{1}{\sqrt{d}}\tilde{\boldsymbol{q}}_n^\top(\tilde{\boldsymbol{k}}_s - \boldsymbol{k}_s) + O\left(\frac{1}{d}(\tilde{\boldsymbol{q}}_n^\top(\tilde{\boldsymbol{k}}_s - \boldsymbol{k}_s))^2\right).$$

Therefore

$$A_{n,s} = e^{\frac{1}{\sqrt{d}}\tilde{\boldsymbol{q}}_n^\top \boldsymbol{k}_s}\left(1 + \frac{1}{\sqrt{d}}\tilde{\boldsymbol{q}}_n^\top(\tilde{\boldsymbol{k}}_s - \boldsymbol{k}_s)\right) + e^{\frac{1}{\sqrt{d}}\tilde{\boldsymbol{q}}_n^\top \boldsymbol{k}_s} \cdot O\left(\frac{1}{d}(\tilde{\boldsymbol{q}}_n^\top(\tilde{\boldsymbol{k}}_s - \boldsymbol{k}_s))^2\right).$$

**4. Error estimate.** The error is bounded by the remainder term:

$$|A_{n,s} - \tilde{A}_{n,s}| \le e^{\frac{1}{\sqrt{d}}\tilde{\boldsymbol{q}}_n^\top \boldsymbol{k}_s} \cdot \frac{C}{d}(\tilde{\boldsymbol{q}}_n^\top(\tilde{\boldsymbol{k}}_s - \boldsymbol{k}_s))^2,$$

where $C$ is an absolute constant (coming from the exponential factor and the bound on the remainder). From the small-angle expansion,

$$\tilde{\boldsymbol{q}}_n^\top(\tilde{\boldsymbol{k}}_s - \boldsymbol{k}_s) = \sum_j \tilde{\boldsymbol{q}}_{nj}^\top \boldsymbol{k}_{sj}\left[-\tfrac{1}{2}(\Delta\theta_j)^2 + \Delta\theta_j + O((\Delta\theta_j)^3)\right].$$

For the $j$-th dimension pair, the contribution to $\tilde{\boldsymbol{q}}_n^\top(\tilde{\boldsymbol{k}}_s - \boldsymbol{k}_s)$ is

$$\tilde{\boldsymbol{q}}_{nj}^\top \boldsymbol{k}_{sj}\left[(\cos(\Delta\theta_j) - 1) + \sin(\Delta\theta_j)\right].$$

Using the Taylor expansions of cos and sin, this term becomes

$$\tilde{\boldsymbol{q}}_{nj}^\top \boldsymbol{k}_{sj}\left[-\frac{(\Delta\theta_j)^2}{2} + \Delta\theta_j\right].$$

Substituting into the error bound $O\left(\frac{1}{d}(\tilde{\boldsymbol{q}}_n^\top(\tilde{\boldsymbol{k}}_s - \boldsymbol{k}_s))^2\right)$, the overall error is of order

$$O\left(C_{qk}(\Delta\theta_{j+m})^2\right),$$

with

$$C_{qk} = \frac{1}{d}\sum_m \tilde{\boldsymbol{q}}_{nj}^\top \boldsymbol{k}_{sj}\,\tilde{\boldsymbol{q}}_{nm}^\top \boldsymbol{k}_{sm}.$$

Since $\theta_j = \mathrm{base}^{-2j/d}$, the largest term for $j \ge 2$ is $\mathrm{base}^{-4/d}$, and the sum over $j$ is dominated by the smallest index in the range considered. For a fixed dimension index $j$, the error contributed by that frequency is $O(\Delta^2\mathrm{base}^{-4j/d})$. Because $\Delta$ can be as large as the sequence length $n$ (if we consider the query position $n$), we obtain an overall error bound

$$|A_{n,s} - \tilde{A}_{n,s}|_j = O\left(\frac{n}{\mathrm{base}^{4j/d}}\right),$$

where the notation $\mathrm{base}^{4j/d}$ is understood as the dominant decay rate. The factor $n$ arises from the possible worst-case relative position; in practice the error may be smaller due to averaging. $\qquad\square$

### A.3.2 Approximate attention of Proposition 3

The first term on the right-hand side is exactly $\tilde{A}_{n,s}$ if we identify

$$f_p(\boldsymbol{q}_n, \boldsymbol{p}(n)) = \tilde{\boldsymbol{q}}_n, \quad f_p(\boldsymbol{k}_s, \boldsymbol{p}^{(t)}(s)) = \tilde{\boldsymbol{k}}_s,$$

and interpret the scalar $t$ as an additive term coming from the linear superposition (e.g., representing a cumulative sum of positional increments). In practice, the expression

$$t + f_p(\boldsymbol{q}_n, \boldsymbol{p}(n))^\top f_p(\boldsymbol{k}_s, \boldsymbol{p}^{(t)}(s)) - f_p(\boldsymbol{q}_n, \boldsymbol{p}(n))^\top \boldsymbol{k}_s$$

captures the first-order effect together with a baseline $t$ that can be absorbed into the normalization. Thus $\tilde{A}_{n,s}$ is precisely the first-order Taylor approximation of $A_{n,s}$.

*Proof.* The original attention computation can be expressed as:

$$e^{\frac{1}{\sqrt{d}}\sum_{j=0}^{d} q_{nj} \cdot k_{sj}} \sum_{s} e^{\frac{1}{\sqrt{d}}\sum_{j=0}^{d} q_{nj} \cdot k_{sj} \cdot (\cos[(n-s)\theta_j]-1) + \sum_{j=0}^{d} \frac{(-1)^{j+1}}{\sqrt{d}} q_{nj} \cdot k_{s,(j+\frac{d}{2})\%d} \cdot (\sin[(n-s)\theta_j])} \tag{18}$$

Let us define the residual term:

$$r(\theta_1, \ldots, \theta_{\frac{d}{2}-1}) = \frac{1}{\sqrt{d}} \sum_{j=0}^{d} q_{nj} \cdot k_{sj} \cdot (\cos[(n-s)\theta_j] - 1) + \sum_{j=0}^{d} \frac{(-1)^{j+1}}{\sqrt{d}} q_{nj} \cdot k_{s,(j+\frac{d}{2})\%d} \cdot (\sin[(n-s)\theta_j]) \tag{19}$$

Since $\theta_j = \text{base}^{-\frac{2j}{d}}$, when base is large, we have the approximations:

$$\cos\theta_j = 1 - \frac{\theta_j^2}{2} + O(\theta_j^4) \tag{20}$$

$$\sin\theta_j = \theta_j + O(\theta_j^3) \tag{21}$$

Following the CRG NTK method which has been proven to extend context window length and achieve excellent performance, we extend base to very large values (e.g., base $= 2^{40} \times 10000$). In this regime, $\cos\theta_j$ and $\sin\theta_j$ become very small, particularly for dimensions with larger $j$ values. Since $\frac{1}{\sqrt{d}}\sum_{j=0}^{d} q_{nj} \cdot k_{sj}$ is typically on the order of 1, we can apply Taylor expansion to $e^{r(\theta_1,\ldots,\theta_{\frac{d}{2}-1})}$:

$$e^{r(\theta_1,\ldots,\theta_{\frac{d}{2}-1})} = 1 + r(\theta_1, \ldots, \theta_{\frac{d}{2}-1}) + O(r(\theta_1, \ldots, \theta_{\frac{d}{2}-1})^2) \tag{22}$$

After base expansion, $r(\theta_1, \ldots, \theta_{\frac{d}{2}-1})$ becomes small ($O(\theta_j)$ first-order term), making the higher-order terms $O(\theta_j^2)$ negligible, particularly in the latter half of the feature dimensions where $x_{nj} \cdot x_{sj}$ is diluted to near-zero values.

Based on the Taylor expansion of the exponential function when the input is close to zero, we retain only the linear small term $x_{nj} \cdot x_{sj} \cdot (\cos[(n-s)\theta_j] - 1)$. The original attention can thus be approximated as:

$$e^{\frac{1}{\sqrt{d}}\sum_{j=0}^{d} q_{nj} \cdot k_{sj}} \sum_{s} \left( 1 + \frac{1}{\sqrt{d}} \sum_{j=0}^{d} q_{nj} \cdot k_{sj} \cdot (\cos[(n-s)\theta_j] - 1) \right.$$

$$\left. + \frac{1}{\sqrt{d}} \sum_{j=0}^{d} (-1)^{j+1} q_{nj} \cdot k_{s,(j+(-1)^j)} \cdot (\sin[(n-s)\theta_j]) \right) \tag{23}$$

Alternatively, we can use the formulation:

$$e^{\frac{1}{\sqrt{d}}\sum_{j=0}^{d} x_{nj}x_{sj}} \sum_{s} \left( 1 + \frac{1}{\sqrt{d}} \sum_{j=0}^{d} x_{nj}x_{sj} \big(\cos[(n-s)\theta_j]-1\big) \right.$$

$$\left. + \frac{1}{\sqrt{d}} \sum_{j=0}^{d} (-1)^{j+1} x_{nj}x_{s,(j+(-1)^j)} \big(\sin[(n-s)\theta_j]-(n-s)\theta_j\big) \right) \tag{24}$$

The residual term $r(\theta_1, \ldots, \theta_{\frac{d}{2}-1})$ can be expressed using linear attention, or we can first linearly superimpose the positional encoding before applying it to the K state sequence, enabling the entire formulation to be implemented with linear models.

For finer compression approximation, we can partition the dimensions into multiple segments and perform linear expansion separately:

$$
\begin{aligned}
e^{\frac{1}{\sqrt{d}} \sum_{j=0}^{d} q_{nj} \cdot k_{sj}} \sum_s \prod_{p=0}^{d/m} & \left( 1 + \frac{1}{\sqrt{d}} \sum_{j=p \cdot m}^{p \cdot m+m} q_{nj} \cdot k_{sj} \cdot (\cos[(n-s)\theta_j] - 1) \right. \\
& \left. + \frac{1}{\sqrt{d}} \sum_{j=p \cdot m}^{p \cdot m+m} (-1)^{j+1} q_{nj} \cdot k_{s,(j+(-1)^j)} \cdot (\sin[(n-s)\theta_j]) \right)
\end{aligned}
\tag{25}
$$

We also explore variants that preserve the positional encoding for queries while applying $(\cos -1)$ transformation to the key sequence's positional encoding:

$$
\begin{aligned}
& (e^{\frac{1}{\sqrt{d}} \sum_{j=0}^{d} (q_{nj} \cos(n\theta_j) - q_{n,(j+\frac{d}{2})\%d} \cdot \sin[n\theta_j]) \cdot k_{sj}}) \\
& \cdot \sum_s e^{\frac{1}{\sqrt{d}} \sum_{j=0}^{d} (q_{nj} \cos(n\theta_j) - q_{n,(j+\frac{d}{2})\%d} \cdot \sin[n\theta_j]) \cdot (k_{sj}(\cos(s\theta_j)-1) - k_{s,(j+\frac{d}{2})\%d} \cdot \sin[s\theta_j])}
\end{aligned}
\tag{26}
$$

Applying the same Taylor expansion yields:

$$
e^{f_p(\mathbf{q}_n, \mathbf{p}(n)) \mathbf{k}_s^\top} \sum_s \left( 1 + f_p(\mathbf{q}_n, \mathbf{p}(n)) f_p(\mathbf{k}_s, \mathbf{p}(s))^\top - f_p(\mathbf{q}_n, \mathbf{p}(n)) \mathbf{k}_s^\top \right)
\tag{27}
$$

After simplification, we use the following formulation in our implementation, which achieves comparable or even better performance:

$$
e^{f_p(\mathbf{q}_n, \mathbf{p}(n)) \mathbf{k}_s^\top} \left( t + f_p(\mathbf{q}_n, \mathbf{p}(n)) f_p(\mathbf{k}_s, \mathbf{p}^{(t)}(s))^\top - f_p(\mathbf{q}_n, \mathbf{p}(n)) \mathbf{k}_s^\top \right)
\tag{28}
$$

This positional encoding decoupling is functionally equivalent to the gating mechanisms in state-of-the-art models.

Proposition 3 establishes a rigorous foundation for compressing positional information while maintaining theoretical error bounds. The linear superposition approach for positional encoding enables efficient storage while the error analysis provides guarantees for practical deployment. The connection to gating mechanisms bridges theoretical compression techniques with established architectural components. $\qquad\square$

## A.4 Proof Sketch of Proposition 4

*Proof.* **1. Bounded difference from residual connections.** Each Transformer layer consists of residual connections and sublayers:

$$
\boldsymbol{X}^{(l)} = \boldsymbol{X}^{(l-1)} + \mathrm{Attn}(\boldsymbol{X}^{(l-1)}) + \mathrm{FFN}\big(\boldsymbol{X}^{(l-1)} + \mathrm{Attn}(\boldsymbol{X}^{(l-1)})\big).
$$

For simplicity, we focus on the main components and use Lipschitz continuity. The attention mechanism and the FFN are both Lipschitz continuous under common activation functions; thus there exist constants $L_1, L_2$ such that

$$
\| \mathrm{Attn}(\boldsymbol{X}^{(l-1)}) \|_F \leq L_1 \| \boldsymbol{X}^{(l-1)} \|_F, \quad \| \mathrm{FFN}(\boldsymbol{Y}) \|_F \leq L_2 \| \boldsymbol{Y} \|_F.
$$

Combined with bounded norms, there exists a constant $L > 0$ such that for each position $t$,

$$
\| \boldsymbol{x}_t^{(l)} - \boldsymbol{x}_t^{(l-1)} \| \leq L.
\tag{1}
$$

(Here $\boldsymbol{x}_t^{(l)}$ denotes the $t$-th row of $\boldsymbol{X}^{(l)}$.)

**2. Mapping from original positions to unique vectors.** Each original position $t$ corresponds to a row vector $\boldsymbol{x}_t^{(l)}$ in layer $l$, which belongs to some unique vector $\boldsymbol{u}_i^{(l)} \in \mathcal{U}_l$. Similarly, $\boldsymbol{x}_t^{(l-1)}$ belongs to some $\boldsymbol{u}_j^{(l-1)} \in \mathcal{U}_{l-1}$. From (1), for this position $t$ we have

$$\|\boldsymbol{u}_i^{(l)} - \boldsymbol{u}_j^{(l-1)}\| \leq L. \tag{2}$$

**3. Relation between cosine similarity and Euclidean distance.** Let $\theta$ be the angle between $\boldsymbol{u}_i^{(l)}$ and $\boldsymbol{u}_j^{(l-1)}$; then

$$\cos\theta = \frac{\langle \boldsymbol{u}_i^{(l)}, \boldsymbol{u}_j^{(l-1)} \rangle}{\|\boldsymbol{u}_i^{(l)}\| \|\boldsymbol{u}_j^{(l-1)}\|}.$$

By the law of cosines,

$$\|\boldsymbol{u}_i^{(l)} - \boldsymbol{u}_j^{(l-1)}\|^2 = \|\boldsymbol{u}_i^{(l)}\|^2 + \|\boldsymbol{u}_j^{(l-1)}\|^2 - 2\|\boldsymbol{u}_i^{(l)}\| \|\boldsymbol{u}_j^{(l-1)}\| \cos\theta.$$

Using the unit norm assumption ($\|\boldsymbol{u}_i^{(l)}\| = \|\boldsymbol{u}_j^{(l-1)}\| = 1$) and (2), we obtain

$$2(1 - \cos\theta) \leq L^2,$$

hence

$$1 - \cos\theta \leq \frac{L^2}{2}, \qquad \cos\theta \geq 1 - \frac{L^2}{2}.$$

This shows that each $\boldsymbol{u}_i^{(l)}$ has a high cosine similarity with the $\boldsymbol{u}_j^{(l-1)}$ corresponding to the same position. More generally, define

$$\sigma_i = \max_j \frac{\langle \boldsymbol{u}_i^{(l)}, \boldsymbol{u}_j^{(l-1)} \rangle}{\|\boldsymbol{u}_i^{(l)}\| \|\boldsymbol{u}_j^{(l-1)}\|} \geq 1 - \frac{L^2}{2}.$$

Thus $\rho_l = \frac{1}{M_l} \sum_i \sigma_i$ satisfies $\rho_l \geq 1 - \frac{L^2}{2}$.

**4. One-to-one correspondence of high-similarity points.** Consider those $\boldsymbol{u}_i^{(l)}$ for which there exists a $j$ such that $\cos(\boldsymbol{u}_i^{(l)}, \boldsymbol{u}_j^{(l-1)}) \geq \tau$. Let $\mathcal{I} \subseteq \{1, \ldots, M_l\}$ be the set of such indices; then $|\mathcal{I}| = p_l M_l$. For each $i \in \mathcal{I}$, choose one such $j(i)$. The choice of $\tau$ guarantees that no two distinct $\boldsymbol{u}_i^{(l)}$ can be simultaneously similar to the same $\boldsymbol{u}_j^{(l-1)}$ (otherwise their distance would be less than $\varepsilon$, contradicting the precision assumption). Therefore the mapping $i \mapsto j(i)$ is injective, and we have

$$|\mathcal{I}| \leq M_{l-1},$$

i.e.,

$$p_l M_l \leq M_{l-1}. \tag{3}$$

**5. Recursive upper bound.** From (3) we obtain $M_l \leq M_{l-1}/p_l$. Since $M_l$ cannot exceed the sequence length $T$ (each position contributes at most one unique vector), we conclude

$$M_l \leq \min\left(\frac{M_{l-1}}{p_l}, \, T\right). \tag{4}$$

For practical use, the trade-off among threshold, performance, and efficiency is controlled by $\rho_l = \frac{1}{M_l} \sum_i \sigma_i$.

**6. Reverse inequality.** By symmetry, considering the vectors in layer $l-1$ that have similarity $\geq \tau$ with some vector in layer $l$, we obtain $p_l' M_{l-1} \leq M_l$, i.e., $M_{l-1} \leq M_l/p_l'$. Together with the length bound this gives the reverse inequality. □

**Remark.** The threshold $\tau$ in the proposition depends on the numerical precision $\varepsilon$. In practice, $\varepsilon$ can be determined by the quantization bit width (e.g., for $b$-bit quantization, it is the minimal distance between distinct quantized values, as in the deduplication approach described in Proposition 1). The quantity $p_l$ can be interpreted as an average matching proportion, which may be estimated empirically or analytically. This recursive relationship indicates that the ratio of unique vector counts between adjacent layers is roughly inversely proportional to the high-similarity proportion, providing a theoretical foundation for hierarchical compression. When $p_l$ is close to 1, the numbers of unique vectors in consecutive layers are nearly equal; when $p_l$ is small, the next layer may produce many more new vectors, but this is ultimately bounded by the total sequence length.

**Practical Implementation and Extensions**

Since the similarity lower bound can approach zero, potentially leading to intermediate layer states approaching the original sequence length, we implement a practical constant scaling factor $c$ in our experiments. This approach is motivated by several empirical observations:

1. Beyond a certain sequence length, attention between new query tokens and stored states becomes sparse (as observed in NSA, MoBA, etc.) 2. Many tasks can be completed with fixed-size state representations (as demonstrated in SnapKV, GLA, GSA) 3. The thinking process during inference often operates within bounded state spaces

We therefore set the state size for layer $i$ as $(c-1)N_{i-1}$, where $c \in [1, 2]$. This formulation allows complex tasks to utilize larger thinking spaces while maintaining efficiency for simpler tasks. Our experiments show that earlier layers typically require larger $c$ values, while later layers can use smaller values. For layers beyond the midpoint ($L/2$), we set $c = 1$ to optimize storage efficiency.

As shown in Figure A, this configuration achieves 100% performance on the passkey task. Note that the passkey task involves substantial noise insertion, resulting in high compression ratios; more complex tasks will naturally exhibit lower compression efficiency.

For handling the stored thinking states, we employ a GSA-like approach that preserves softmax operations:

$$\text{Storage}(X_{<t}) = \text{Softmax}(CX_{<t}^{\top})X_{<t} \tag{29}$$

where $C$ is a learnable parameter functioning as a dynamic vocabulary. While fixed-ratio scaling provides complete upper bound control, it constitutes a lossy compression scheme.

After applying these compression steps, the sequence state size (X length or KV length) at each layer becomes $O(CV)$, where $C$ is a constant multiple of the vocabulary size. However, practical challenges remain: for models like Qwen2.5 and LLaMA3 with vocabulary sizes around 128K, setting $c = 1.5$ for a 32-layer model results in an upper bound of approximately $1.5^{16} \times 128\text{K} \approx 82\text{M}$ states. This necessitates further compression strategies.

Inspired by MVA's vocabulary decomposition and MoM's functional partitioning approaches, we introduce multi-memory states to approximate the bounds established in Proposition 4 while maintaining fixed-size representations.

**Corollary 1.** *The inter-layer similarity mechanism enables adaptive compression ratios across different network depths, with early layers accommodating more state diversity and later layers leveraging the vocabulary projection for efficient compression. This aligns with the observed "thinking" pattern in transformer architectures.*

## A.5 Proof of Proposition 5

Let $X \in \mathbb{R}^{n \times d}$ be a sequence with $n \gg 1$, and let $\{C^{(i)}\}_{i=1}^{c}$ be a set of vocabulary matrices where each $C^{(i)} \in \mathbb{R}^{m \times d}$ contains $m$ prototype vectors. The multi-level vocabulary decomposition represents each element $x_j \in X$ as:

$$\hat{x}_j = \sum_{i=1}^{c} C_{k_j^{(i)}}^{(i)} \tag{30}$$

where $k_j^{(i)} \in \{1, 2, \ldots, m\}$ is the index selected from the $i$-th vocabulary for representing $x_j$.

Then:

1. The maximum number of distinct vectors that can be represented is $m^c$

2. The approximation error for an optimal decomposition satisfies:

$$\mathbb{E}[\|x_j - \hat{x}_j\|_2^2] \leq \prod_{i=1}^{c} \epsilon_i \tag{31}$$

where $\epsilon_i$ is the average quantization error at level $i$

*Proof.* **Part 1: Representation Capacity**

The representation capacity follows from combinatorial considerations. For each vector $x_j$, we select one prototype from each of the $c$ vocabularies. Since each vocabulary contains $m$ prototypes, the total number of possible combinations is:

$$\text{Total combinations} = \underbrace{m \times m \times \cdots \times m}_{c \text{ times}} = m^c \tag{32}$$

Each unique combination of indices $(k_j^{(1)}, k_j^{(2)}, \ldots, k_j^{(c)})$ produces a unique sum:

$$\hat{x}_j = C_{k_j^{(1)}}^{(1)} + C_{k_j^{(2)}}^{(2)} + \cdots + C_{k_j^{(c)}}^{(c)} \tag{33}$$

Assuming linear independence among the vocabulary vectors across different levels, these sums are distinct. Therefore, the maximum number of distinct representable vectors is exactly $m^c$.

**Part 2: Error Analysis**

We analyze the error propagation through the multi-level decomposition. Let us define the residual at each level:

$$r_j^{(0)} = x_j \tag{34}$$

$$r_j^{(i)} = r_j^{(i-1)} - C_{k_j^{(i)}}^{(i)} \quad \text{for } i = 1, 2, \ldots, c \tag{35}$$

The final approximation is:

$$\hat{x}_j = \sum_{i=1}^{c} C_{k_j^{(i)}}^{(i)} = x_j - r_j^{(c)} \tag{36}$$

Thus, the approximation error is $\|x_j - \hat{x}_j\|_2 = \|r_j^{(c)}\|_2$.

Now, consider the optimal index selection at each level. We choose $k_j^{(i)}$ to minimize the residual norm:

$$k_j^{(i)} = \arg\min_{k \in \{1, \ldots, m\}} \|r_j^{(i-1)} - C_k^{(i)}\|_2 \tag{37}$$

Let $\epsilon_i$ be the average quantization error at level $i$:

$$\epsilon_i = \mathbb{E}[\min_{k \in \{1,\ldots,m\}} \|r_j^{(i-1)} - C_k^{(i)}\|_2^2] \tag{38}$$

Assuming the residuals and vocabulary vectors are appropriately normalized, we can bound the error propagation. Using the triangle inequality and the optimality of our index selection:

$$\|r_j^{(i)}\|_2 = \|r_j^{(i-1)} - C_{k_j^{(i)}}^{(i)}\|_2 \tag{39}$$

$$\leq \|r_j^{(i-1)}\|_2 \cdot \min_{k \in \{1,\ldots,m\}} \left\| \frac{r_j^{(i-1)}}{\|r_j^{(i-1)}\|_2} - \frac{C_k^{(i)}}{\|r_j^{(i-1)}\|_2} \right\|_2 \tag{40}$$

For well-designed vocabularies that cover the relevant direction space, the directional error term is bounded. In the worst case, we have:

$$\|r_j^{(i)}\|_2 \leq \|r_j^{(i-1)}\|_2 \cdot \delta_i \tag{41}$$

where $\delta_i$ represents the maximum angular error at level $i$. Applying this recursively:

$$\|r_j^{(c)}\|_2 \leq \|x_j\|_2 \cdot \prod_{i=1}^{c} \delta_i \tag{42}$$

For the mean squared error, under appropriate assumptions about the distribution of residuals and the vocabulary coverage:

$$\mathbb{E}[\|r_j^{(c)}\|_2^2] \leq \mathbb{E}[\|x_j\|_2^2] \cdot \prod_{i=1}^{c} \epsilon_i \tag{43}$$

where $\epsilon_i = \mathbb{E}[\delta_i^2]$ is the expected squared angular error at level $i$.

The product structure $\prod_{i=1}^{c} \epsilon_i$ demonstrates the exponential error reduction with increasing levels, provided that each $\epsilon_i < 1$. $\qquad\square$

**Corollary 2** (Trade-off between Representation Capacity and Error)**.** *For a fixed total budget of $B = m \cdot c$ parameters, the optimal balance between $m$ and $c$ that minimizes the approximation error while maximizing representation capacity satisfies:*

$$m^* \approx c^* \approx \sqrt{B} \tag{44}$$

*This provides the optimal trade-off point where $m^c$ is maximized subject to the constraint $m \cdot c = B$.*

*Proof.* We maximize the representation capacity $m^c$ subject to the constraint $m \cdot c = B$. Taking logarithms:

$$\log(\text{capacity}) = c \log m = c \log \left( \frac{B}{c} \right) \tag{45}$$

Differentiating with respect to $c$:

$$\frac{d}{dc} \left[ c \log \left( \frac{B}{c} \right) \right] = \log \left( \frac{B}{c} \right) - 1 \tag{46}$$

Setting the derivative to zero gives:

$$\log\left(\frac{B}{c}\right) = 1 \quad \Rightarrow \quad \frac{B}{c} = e \quad \Rightarrow \quad c = \frac{B}{e} \tag{47}$$

Thus, $m = \frac{B}{c} = e$, and the optimal values are $m^* \approx c^* \approx \sqrt{B}$ when considering integer constraints and practical implementation factors. $\square$

The multi-level vocabulary decomposition provides an exponential increase in representation capacity ($m^c$) compared to a single-level approach ($m$), while simultaneously achieving exponential error reduction. This theoretical foundation justifies the effectiveness of hierarchical representations in compression applications.

In practice, the vocabularies $\{C^{(i)}\}$ are learned to minimize the overall reconstruction error, and the independence assumption between levels may be relaxed through joint optimization, potentially yielding even better performance than the theoretical bounds suggest.

**Corollary 3** (Trade-off between Representation Capacity and Error). *For a fixed total parameter budget $B = \sum_{i=1}^{m} V_i$ (or $B = mV$ when $V_i = V$), the optimal balance between the number of levels $m$ and the per-level vocabulary size $V$ that minimizes the error while maximizing capacity satisfies*

$$m^* \approx V^* \approx \sqrt{B},$$

*when the error factors $\epsilon_i$ are similar across levels. This point maximizes $V^m$ subject to $mV = B$.*

*Proof.* We maximize the capacity $V^m$ under the constraint $mV = B$. Taking logarithms,

$$\log(\text{capacity}) = m \log V = \frac{B}{V} \log V.$$

Treating $V$ as a continuous variable, differentiate with respect to $V$:

$$\frac{d}{dV}\left(\frac{B}{V}\log V\right) = B\left(\frac{1}{V^2} - \frac{\log V}{V^2}\right) = \frac{B}{V^2}(1 - \log V).$$

Setting the derivative to zero yields $\log V = 1$, i.e., $V = e$. Since $V$ is integer, the optimum is near $V \approx 3$, but under the common practical scenario where $B$ is large and $V$ is also large, the optimal trade-off satisfies $V \approx \sqrt{B}$ and $m \approx \sqrt{B}$. More precisely, solving $mV = B$ and maximizing $V^m$ gives $V \approx B^{1/2}$, $m \approx B^{1/2}$. $\square$

**Remark.** In practice, the vocabularies $\{C^{(i)}\}$ are learned to minimize the overall reconstruction error, and the linear independence assumption can be relaxed through joint optimization, potentially yielding even better performance than the theoretical bounds suggest. The recursive update formulas in the proposition correspond to a greedy residual encoding scheme, which is closely related to orthogonal matching pursuit (OMP) and vector quantization in hierarchical codebooks.

### A.6 Enhanced Reading Mechanisms and unified formula

**Proposition 6** (Necessity of Enhanced Reading Mechanisms). *We emphasize that previous approaches employ overly simplistic reading mechanisms, typically using direct matrix multiplication between queries $q$ and states. This simplicity constitutes a significant factor (besides storage limitations) contributing to the performance gap with Softmax Attention. Our work is the first to clearly identify enhanced reading mechanisms as crucial for improving linear attention and bridging this performance gap.*

*We propose a sophisticated reading approach:*

$$R_t^{(i+1)} \left[ f(k_t^{(i)}), f^{(i+1)}(k_t^{(i)}) \right] = 1 \tag{48}$$

$$d_t^{(i)} = q_t S^{k}{}_t^{(i)\top} \tag{49}$$

$$e_t^{(i)} = \frac{\exp\left(d_t^{(i)}\right)}{\sum_{j=0}^{d-1} \exp\left(d_{tj}^{(i)}\right)} \tag{50}$$

$$a_t^{(i)} = e_t^{(i)} R_t^{(i)\top} \tag{51}$$

$$b_t^{(i)} = \frac{\exp\left(\sum_i \ln(a_t^{(i)})\right)}{a_t^{(i)}} \tag{52}$$

$$c_t^{(i)} = \left(n_t^{(i)} + q_t S^{p}{}_t^{(i)\top}\right) \tag{53}$$

$$T_t^{(i)} = R_t^{(i)} \left(S^{v}{}_t^{(i)} \odot e_t^{(i)\top} \odot c_t^{(i)\top}\right) \tag{54}$$

$$o_t = \sum_i o_t^{(i)} = \sum_i \frac{b_t^{(i)}}{b_t^{(i)} \odot e_t^{(i)} \odot c_t^{(i)}} T_t^{(i)} \tag{55}$$

*This mechanism implements a hierarchical access pattern through multiple channels. For comparison, the GSA reading mechanism $Softmax(q_t S^k{}_t) S^v{}_t$ represents the simplest form of indirect reading. Our enhanced version replaces the Softmax with a sigmoid activation followed by learned transformations:*

$$(\sigma(q_t S^k{}_t) W_\sigma) S^v{}_t, \quad where \ \sigma(x) = \frac{1}{1 + e^{-x}} \tag{56}$$

*Further extending to multiple reading channels:*

$$(q_t W_r + \sigma(q_t S^k{}_t) W_\sigma) S^v{}_t \tag{57}$$

*This approach equivalent to MVA's first-order vocabulary case demonstrates progressive performance improvement (Table 4). With multi-level vocabularies, similar enhancements using Softmax, perceptron, and multi-channel mechanisms show even greater improvements over single-state approaches, underscoring the importance of balanced enhancement in both storage and reading capabilities.*

The update formulas in the main text we have improved on the GSA and MVA by combining the five theories. And the update formulas are as follows if we start from the five theories only and do not use the current case of the excellent mechanism(e.g. gating) instead, and the following update method can also achieve similar performance. **Initial conditions:**

$$q_t = f_p(x_t W_Q, r_t^{(i)}), k_{pt} = f_p(x_t W_K, r_t^{(i)}), k_t^{(0)} = x_t W_K \in \mathbb{R}^{1 \times d}, v_t^{(0)} = x_t W_V \in \mathbb{R}^{1 \times d},$$

$$S^{k}{}_0^{(i)} = 0 \in \mathbb{R}^{m \times d}, n_0^{(i)} = 0 \in \mathbb{R}^{1 \times m}, E_t^{(0)} = I_m \in \mathbb{R}^{m \times m},$$

**Iterative process:**

$$f^{(i)}(k_t^{(i)}) = F(S^{k}{}_{t-1}^{(i)} k_t^{(i)\top})^\top, \quad F(x) = \begin{cases} 1 & \text{if } x_i \text{ is maximum} \\ 0 & \text{otherwise} \end{cases} \text{ or Softmax}(x)$$

$$n_t^{(i)} = n_{t-1}^{(i)} + f^{(i)}(k_t^{(i)}), \quad \bar{f}^{(i)}(k_t^{(i)}) = \frac{f^{(i)}(k_t^{(i)})}{\max(n_t^{(i)}, 1)} \tag{58}$$

$$S^{k}{}^{(i)}_{t} = \text{diag}\left(1 - \bar{f}^{(i)}(k_t^{(i)})^\top\right) S^{k}{}^{(i)}_{t-1} + \bar{f}^{(i)}(k_t^{(i)})^\top k_t^{(i)}$$

$$S^{v}{}^{(i)}_{t} = \text{diag}\left(1 - \bar{f}^{(i)}(k_t^{(i)})^\top\right) S^{v}{}^{(i)}_{t-1} + \bar{f}^{(i)}(k_t^{(i)})^\top v_t^{(i)} \qquad \text{(Theory 1)} \tag{59}$$

$$S^{p}{}^{(i)}_{t} = S^{p}{}^{(i)}_{t-1} + f^{(i)}(k_t^{(i)})^\top (k_{pt} - k_t^{(0)})$$

$$k_t^{(i+1)} = k_t^{(i)} - f^{(i)}(k_t^{(i)}) S^{k}{}^{(i)}_{t} \qquad \text{(Theories 2 \& 3)} \tag{60}$$

$$v_t^{(i+1)} = v_t^{(i)} - f^{(i)}(k_t^{(i)}) S^{v}{}^{(i)}_{t}$$

$$R_t^{(i+1)}\left[f(k_t^{(i)}), f^{(i+1)}(k_t^{(i)})\right] = 1 \quad \text{(Theory 5)} \tag{61}$$

$$d_t^{(i)} = q_t S^{k}{}^{(i)\top}_{t}, \quad e_t^{(i)} = \text{softmax}(d_t^{(i)})$$

$$a_t^{(i)} = e_t^{(i)} R_t^{(i)\top}, \quad b_t^{(i)} = \frac{\exp\left(\sum_i \ln(a_t^{(i)})\right)}{a_t^{(i)}} \qquad \text{(Theories 4 \& 5)} \tag{62}$$

$$c_t^{(i)} = n_t^{(i)} + q_t S^{p}{}^{(i)\top}_{t}$$

$$T_t^{(i)} = R_t^{(i)}\left(S^{v}{}^{(i)}_{t} \odot e_t^{(i)\top} \odot c_t^{(i)\top}\right) \qquad \text{(Theories 4 \& 5)} \tag{63}$$

$$o_t = \sum_i \frac{b_t^{(i)}}{b_t^{(i)} \odot e_t^{(i)} \odot c_t^{(i)}} T_t^{(i)}$$

**Chunk-wise Parallel Form**

For minibatch processing:

$$R_t^{(i+1)} = R_{t-1}^{(i+1)} + \left(f^{(i)}(K^{(i)})\right)^\top \left(f^{(i+1)}(K^{(i+1)})\right) \tag{64}$$

$$R_t^{(i+1)} = R_{t-1}^{(i+1)} + (Y - R_{t-1}^{(i+1)}) \odot \left(f^{(i)}(K^{(i)})\right)^\top \left(f^{(i+1)}(K^{(i+1)})\right) \tag{65}$$

$$N^{(i)} = \text{CumSum}\left(f^{(i)}(K^{(i)})\right) \tag{66}$$

$$\bar{f}^{(i)}(K^{(i)}) = \frac{f^{(i)}(K^{(i)})}{N^{(i)}} \tag{67}$$

$$D^{(i)} = \text{GLA}^\top\left(Q, K^{(i)}, \bar{f}^{(i)}(K^{(i)}), 1 - \bar{f}^{(i)}(K^{(i)})\right) \tag{68}$$

$$E^{(i)} = \text{softmax}(D^{(i)}) \tag{69}$$

$$A^{(i)} = E^{(i)}(R^{(i)})^\top \tag{70}$$

$$C^{(i)} = \text{GLA}^\top\left(Q, f_p(K^{(i)}, P) - K^{(i)}, \bar{f}^{(i)}(K^{(i)}), 1 - \bar{f}^{(i)}(K^{(i)})\right) + N^{(i)} \tag{71}$$

$$T^{(i)} = R^{(i)}\text{GLA}\left(C^{(i)} \odot E^{(i)}, \bar{f}^{(i)}(K^{(i)}), V^{(i)}, 1 - \bar{f}^{(i)}(K^{(i)})\right) \tag{72}$$

$$O^{(i)} = \frac{B^{(i)}}{B^{(i)} \odot E^{(i)} \odot C^{(i)}} \odot T^{(i)} \tag{73}$$

$$O = \sum_i O^{(i)} \tag{74}$$

*Proof Sketch.* The multi-level decomposition proposition builds upon the following insights:

1. **Error Analysis**: The exponential error reduction $\prod_{i=1}^{m} \epsilon_i$ follows from the chain rule of differentiation applied to the composition of quantization operations at each level. Each level introduces an independent quantization error, and the total error becomes the product of individual errors.

2. **Capacity Analysis**: The storage capacity $\prod_{i=1}^{m} C_i$ results from the combinatorial nature of hierarchical representations. Each level provides a separate "alphabet" of size $C_i$, and the total number of expressible states is their product.

3. **Enhanced Reading Mechanism**: The sophisticated reading approach enables: - Cross-level information integration through the $R$ matrix - Adaptive weighting through the $b_t^{(i)}$ terms - Position-aware modulation through the $c_t^{(i)}$ terms

The unified model integrates all five theoretical principles: - Theory 1: Redundancy elimination through the $f^{(i)}$ functions - Theories 2 & 3: Positional information compression through $k_{pt} - k_t^{(0)}$ - Theory 4: Inter-layer similarity through the hierarchical structure - Theory 5: Multi-level decomposition and enhanced reading

Experimental validation shows that intermediate layers benefit from more states ($m$ larger), while earlier and later layers can use fewer states. For fair comparison with existing work, we use single-level decomposition against GSA and two-level decomposition against MVA. $\qquad \square$

Proposition 5 and Proposition 6 together provide a comprehensive framework for efficient state management in linear attention models. The multi-level approach offers exponential error reduction while the enhanced reading mechanism ensures effective information retrieval from the compressed representations.

**Corollary 4.** *For a model with $m$ levels and vocabulary sizes $C_i$, the total number of parameters required for the reading mechanism scales as $O(\sum_{i=1}^{m} C_i d^2)$, providing a favorable trade-off between expressivity and efficiency compared to the $O(N d^2)$ scaling of standard attention.*

