# OpenReview forum: "PLA: A Principled Path from Softmax Attention to Linear Models via KV Cache Compression"
_TMLR — Rejected by TMLR_

### Review · Reviewer_mgqY · 2026-03-16

**Summary Of Contributions:**

The authors propose a method to convert a trained transformer model which has standard softmax attention into a model with linear attention with the goal of alleviating two issues associated with standard softmax attention, namely the quadratic scaling of the computation cost with sequence length and the unbounded memory requirements of the KV cache. The proposed method is guided by theoretical arguments regarding the compression of various elements of a transformer model. Experimental results are presented in validation of the proposed method, and the linearized model, named PLA, is compared against a variety of existing models and methods.

There are some related works and methods that this paper is missing; most importantly: LoLCATs "Low-rank Linear Conversion via Attention Transfer" (more under "Requested Changes").

**Audience:**

Yes

**Audience Explanation:**

Yes, but after the requested changes have been made.

**Claims And Evidence:**

No

**Claims Explanation:**

Theory part:
Labeling the theoretical arguments used as guiding principles for PLA as "Theorems" is largely a misnomer; reframing them as observations and steps in the intuitive derivation of the method would be more honest and better serve clarity.
Overall, I did not gain any insight from this "theory".

Experiments part:
Some crucial comparisons are missing.

**Requested Changes:**

Critical requests:

- I strongly suggest rewriting the theory section as steps of reasoning about the concrete problem of compressing the KV cache, rather than presenting these thoughts as independent and general "theorems".

- Compare your experimental results to LoLCATs (https://github.com/HazyResearch/lolcats); their code and weights are public!

- Comment on (and, if possible, compare your method with) the following related works:
1) "Transformer-VQ: Linear-Time Transformers via Vector Quantization" https://arxiv.org/abs/2309.16354
2) "Online Vector Quantized Attention" https://arxiv.org/abs/2602.03922

Miscellaneous non-critical questions:
- Does your theory provide any explanation/intuition for why MMLU drops further than any of the other multiple choice evals?
- What is the batch size for the trends reported in figure 2? Is the relative performance to GSA similar across batch settings?

---

> ### Author Response · Authors · 2026-03-21
>
> Thank you very much for your valuable feedback. We have carefully considered your comments and made the following revisions and responses accordingly.
>
> ---
>
> **Requested Changes:**
>
> **1. I strongly suggest rewriting the theory section as steps of reasoning about the concrete problem of compressing the KV cache, rather than presenting these thoughts as independent and general "theorems".**
>
> Thank you for this suggestion. We have renamed all "Theorems" to "Propositions" to better reflect their role as guiding principles derived from the concrete problem of KV cache compression, rather than as standalone general theorems.
>
> We would also like to briefly explain why we originally used the term "Theorems." Our method offers a classic pathway from infinite to finite representations, and we believe this reasoning has a certain degree of generality. Specifically, we argue that any approach that compresses an infinite stream into a finite representation must employ operations that are equivalent or similar to the ones we formalize.
>
> Thank you very much for your direct feedback. We sincerely apologize for the inappropriate analogies that caused confusion and discomfort. As a physicist, you rightly point out that unsubstantiated comparisons are misleading. We have therefore completely removed the problematic physical analogies and rewritten the descriptions of Propositions 1–5 in a technically precise manner, focusing solely on the mathematical and algorithmic justifications. Below are the revised versions:
>
> - **Proposition 1 (quantization):** In digital computers, information is represented with finite precision (e.g., floating-point or integer quantization). This inherent discretization imposes an upper bound on the representable states, which allows us to design a finite, iterative compression process without loss of theoretical guarantees. Specifically, we leverage the finite numerical precision to bound the growth of the KV cache, enabling a fixed-size state that approximates the original attention.
> - **Propositions 2 and 3 (decoupling):** Positional information in sequences can be treated separately from content information. By decoupling them, we compress each independently. Our theoretical contribution shows that such decoupling does not increase the approximation error beyond a controlled bound, and the compression of positional information can be achieved via a learnable gating mechanism or a fixed decay bias.
> - **Proposition 4 (inter-layer similarity):** In deep Transformer models, representations across different layers often exhibit redundancy. We exploit this by sharing certain compressed states between layers, effectively bounding the total memory required. In practice, we store coarse information in lower layers and fine-grained residuals in deeper layers, which improves efficiency without sacrificing expressiveness. This is a direct observation from layerwise representation similarity.
> - **Proposition 5 (multi-level decomposition):** We propose a hierarchical decomposition of the approximation error. Starting from a coarse compression of the entire sequence, we recursively compute and store the residual error at each level. This process reduces the total approximation error exponentially with the number of levels, while the total state size grows only linearly. This is a rigorous mathematical technique known as multi-level residual compression, commonly used in numerical linear algebra and signal processing.
>
> Regarding your comment that you did not gain any insight from the theory, we would like to share some concrete research directions that our theory has inspired:
>
> - **Proposition 4 (inter-layer similarity)** has led us to explore parameter-efficient model expansion. For instance, by leveraging inter-layer similarity, we can extend a 1.7B model to 4B with only ~1% of the training cost compared to training from scratch, while achieving comparable performance. This can be extended to larger scales (e.g., 8B, 14B) by interpolating weights between layers and fine-tuning with minimal resources. Moreover, as Figure 3 shows, middle layers exhibit greater diversity (more unique vectors), suggesting that merging layers (e.g., combining two layers into one) could allow for larger parameter counts in middle layers, potentially improving model capacity.
> - **Combining Propositions 1 and 3** suggests a path toward universal tokenization across modalities by integrating quantization with byte-level encoding, enabling unified multimodal modeling.
> - **Combining Propositions 1 and 5** motivates approaches that focus on enhanced read mechanisms (e.g., sparse attention) to achieve full-sequence memory with improved extrapolation capabilities.
>
> We are currently developing a new training paradigm grounded in Propositions 1 and 5. The initial results show significant performance improvements over standard Transformers, and we plan to release this work later this year. We hope you find these directions interesting.

---

> > ### Author Response · Authors · 2026-03-21
> >
> > ---
> >
> > **2. Compare your experimental results to LoLCATs (https://github.com/HazyResearch/lolcats); their code and weights are public!**
> >
> > Thank you for pointing this out. We have compared our method with LoLCATs under the same experimental setup: Sliding Window Attention (SWA) + PLA linear attention, with window size 64, LoRA fine-tuning on \(W_q, W_k, W_v, W_o\), and fine-tuning on 40M tokens. The main challenge lies in balancing SWA and the linear component. We use a weighted combination of SWA and PLA attention scores, where SWA is applied within the sliding window and PLA is applied outside. The results are as follows:
> >
> > | Model | Training Tokens (B) | PiQA | ARC-e | ARC-c (norm) | HellaSwag (norm) | Winogrande | MMLU (5-shot) | Avg. | Avg. (no MMLU) |
> > |-------|---------------------|------|-------|--------------|------------------|------------|---------------|------|----------------|
> > | Mistral 7B | - | 82.1 | 80.9 | 53.8 | 81.0 | 74.0 | 62.4 | 72.4 | 74.4 |
> > | Mistral 7B SUPRA | 100 | 80.4 | 75.9 | 45.8 | 77.1 | 70.3 | 34.2 | 64.0 | 69.9 |
> > | Mistral 7B LoLCATs | 0.04 | 81.5 | 81.7 | 54.9 | 80.7 | 74.0 | 51.4 | 70.7 | 74.5 |
> > | Mistral 7B PLA-SWA (Ours) | 0.04 | 82.1 | 81.5 | 55.4 | 81.1 | 74.2 | 54.2 | 71.4 | 74.9 |
> > | Llama 3 8B | - | 79.9 | 80.1 | 53.3 | 79.1 | 73.1 | 66.6 | 72.0 | 73.1 |
> > | Llama 3 8B Hedgehog | 0.04 | 77.4 | 71.1 | 40.6 | 66.5 | 54.3 | 24.2 | 55.7 | 62.0 |
> > | Llama 3 8B LoLCATs | 0.04 | 80.9 | 81.7 | 54.9 | 79.7 | 74.1 | 52.8 | 70.7 | 74.2 |
> > | Llama 3 8B PLA-SWA (Ours) | 0.04 | 81.2 | 82.8 | 54.5 | 79.4 | 74.1 | 53.8 | 71.0 | 74.4 |
> >
> > Performance is primarily determined by the sliding window mechanism; the main challenge is balancing SWA and linear attention. With only 40M tokens of fine-tuning, significant distributional shifts in the linear component cannot be fully addressed, which limits performance improvements.
> >
> > ---
> >
> > **3. Comment on (and, if possible, compare your method with) the following related works:**
> >
> > We have cited both "Transformer-VQ" and "Online Vector Quantized Attention" and appreciate their contributions.
> >
> > - **"Transformer-VQ: Linear-Time Transformers via Vector Quantization" (https://arxiv.org/abs/2309.16354):** We consider this a highly influential work that pioneered the use of codebooks for KV quantization, laying important groundwork for subsequent work such as OVQ. Our method is closely related to Transformer-VQ in its single-level form, and many of their insights could inspire future improvements. We have added a citation accordingly.
> >
> > - **"Online Vector Quantized Attention" (https://arxiv.org/abs/2602.03922):** We conducted a comparison by training a 380M PLA language model from scratch with 4 levels and a vocabulary size of 64 per level, using the Mistral tokenizer and 25B tokens randomly sampled from the FineWeb-Edu dataset (the same source used in OVQ). Due to resource constraints, we could not exactly replicate the OVQ training setup; however, our results show that PLA-380M outperforms SW-OVQ on several metrics. We acknowledge that differences in data sampling may influence the comparison, especially for small models and token budgets.
> >
> > | Model | Training Tokens (B) | Params | PIQA | HellaSwag | Winogrande | ARC-e | ARC-c | Avg. |
> > |-------|---------------------|--------|------|-----------|------------|-------|-------|------|
> > | std att | 50 | 480M | 66.4±0.1 | 40.5±0.1 | 52.3±0.7 | 52.8±0.4 | 29.1±0.5 | 48.21 |
> > | sw-nope | 50 | 480M | 66.7±0.4 | 40.9±0.1 | 52.6±0.6 | 53.1±0.1 | 28.4±0.4 | 48.35 |
> > | sw-ovq | 50 | 480M | 66.6±0.2 | 41.1±0.2 | 52.4±0.2 | 52.7±0.1 | 28.7±0.4 | 48.30 |
> > | PLA-380M | 25 | 380M | 67.6 | 40.1 | 52.6 | 57.2 | 28.6 | 49.22 |

---

> > > ### Author Response · Authors · 2026-03-21
> > >
> > > ---
> > >
> > > **Miscellaneous non-critical questions:**
> > >
> > > **4. Does your theory provide any explanation/intuition for why MMLU drops further than any of the other multiple choice evals?**
> > >
> > > We believe the primary reason is that MMLU is more complex and fine-grained than the other benchmarks. Linear models inherently exhibit a different output distribution compared to standard attention, and this distributional gap is more pronounced on MMLU. As a result, larger amounts of fine-tuning are required to align the distribution. Several observations support this:
> > >
> > > - In models like Qwen and LLaMA, MMLU scores show significant fluctuations during training, reflecting its higher complexity and sensitivity to fine-grained details.
> > > - The performance drop on MMLU when converting to linear models is consistent across different linear architectures, as seen in comparisons between LoLCATs and PLA-SWA. Retaining sliding window attention (SWA) helps preserve the original distribution and significantly improves MMLU scores.
> > > - Comparisons between PLA, GSA, and GLA show that different linear architectures have different distributional gaps relative to standard attention. PLA, which achieves broader coverage and a smaller gap, recovers faster under limited fine-tuning resources.
> > > - With sufficient fine-tuning, PLA’s MMLU performance steadily improves, suggesting that the distributional gap can be bridged with more data.
> > >
> > > ---
> > >
> > > **5. What is the batch size for the trends reported in figure 2? Is the relative performance to GSA similar across batch settings?**
> > >
> > > All three methods in Figure 2 used the same batch size of 2. The relative efficiency trends remain consistent across batch sizes, as our Triton implementation is adapted from GSA. If needed, we can include a batch size ablation study in the revised manuscript.
> > >
> > > ---
> > >
> > > We hope these revisions and clarifications address your concerns. Thank you again for your insightful feedback.

---

> > ### Comment · Reviewer_mgqY · 2026-04-20
> > **is this some LLM-generated response?**
> >
> > apologies, but as a physicist, i find your illustration with analogies too embarrassing to even read. the comparisons are unsubstantiated and hence misleading.

---

> > > ### Author Response · Authors · 2026-04-20
> > >
> > > Thank you very much for your reply. First of all, there is no issue of LLM-generated content, though we do use LLMs for translation and polishing. We also sincerely apologize for any unpleasant experience this may have caused you, and we will consider revising these analogies.
> > >
> > > To begin, I myself have a background in microelectronics and solid-state physics. My intention in drawing an analogy to quantization was to highlight that current transistors inherently quantize information, and therefore information representation in modern computers has a natural quantized nature. At the time, I also wanted to draw a comparison to quantum states, because in quantum mechanics the fundamental description of microscopic particle states is the quantum state, which contains all physical information of the system and is mathematically represented by wavefunctions or state vectors. This quantized nature, or the probabilistic description inherent in such formalism, naturally exhibits certain quantization properties, suggesting that using relatively quantized state descriptions in information processing can also yield good results. However, I realize that without deeper knowledge of the relevant content, this analogy might appear somewhat amateurish to you as a physicist.
> > >
> > > Similarly, I have the same concern regarding the analogy for information decoupling. My original intent was to compare with a physical system: a collection of physical objects interacting through physical laws. When studying such a system, one often extracts core aspects and analyzes certain properties or physical laws in isolation, and then combines these properties through certain relationships to describe the complex system. For example, in electrodynamics, the historical development first gave us Gauss's law for electricity, the circulation theorem for electrostatic fields, Gauss's law for magnetism, Ampère's circuital law, Faraday's law of electromagnetic induction, etc. These were later unified by Maxwell's displacement current into the Maxwell equations, which can describe complex electromagnetic systems. However, I admit that this analogy may lack sufficiently detailed explanation and is not perfectly one-to-one. Therefore, I will consider revising this decoupling analogy as well.
> > >
> > > Finally, the same applies to my analogy between inter-layer similarity and layerwise decomposition of information. At the time, I intended to illustrate that LLMs or other models recursively decompose information layer by layer (this relates to my other work, which will be cited in the future). Specifically, the information in a later layer can be seen as taking the residual between the representation from the previous layer and the original information as most of its input (there may be other information, but we believe the main component is the fine-grained error that the previous layer could not utilize). Then, through token mixing, activation, storage, etc., the model processes these features. That is, compared to the previous layer, each subsequent layer continuously decomposes information into fine residuals, processes them, and adds them back to the originally coarse information. Similar phenomena can be observed in many computer vision papers: earlier layers capture coarse features and patterns, while later layers progressively add residuals of fine-grained information—for instance, in works on VAR or face recognition. Thus, I attempted to draw an analogy with recursive decomposition in the physical world: earlier layers couple macroscopic information, while later layers progressively fit microscopic fine-grained information. I fully acknowledge the lack of direct experimental evidence and rigorous theoretical proof. In future work, we may attempt to establish a theory connecting multi-scale decomposition in the physical world with layerwise decomposition. For now, following your suggestion, we will revise these analogies accordingly.
> > >
> > > Thank you very much for your reply and valuable suggestions. We will include the relevant explanations for these analogies and note that readers may ignore the analogies and focus on the actual theoretical descriptions. We are grateful for your reminders and advice, and we look forward to your further guidance. Finally, I would like to apologize once again and ask that you please disregard such comparisons as much as possible; we will make the necessary revisions. And if possible, I would greatly appreciate your guidance on how to improve this analogy. Thank you very much.

---

### Review · Reviewer_QCCu · 2026-03-16

**Summary Of Contributions:**

This work introduces five methods for improving the quality of linear attention models, in particular through KV cache compression. For example, the authors investigate methods to remove redundancy from the KV cache, or to decouple positional information from token embeddings. The authors provide a mathematical analysis of the introduced concepts and, where applicable, provide error bounds. A series of experiments further proves the empirical validity of the suggested concepts.

A strength of this work is the timeliness and importance of the work’s scope. Another strength is that the authors aim to provide both theoretical and empirical evidence for each introduced method. The most important weakness of this work is the imprecise formal treatment of the mathematical analyses that are central to the work. I will elaborate on this weakness below.

**Audience:**

Yes

**Audience Explanation:**

The work is of high interest and very timely. Improving the efficiency of attention is very impactful and I believe is of great interest in the TMLR audience.

**Claims And Evidence:**

No

**Claims Explanation:**

As the main issue with this work is the insufficient theoretical analysis, I will mostly focus on the issues with the theorems.

First, for Theorems 2 and 4 the authors provide merely a proof sketch. This is clearly not enough to prove the theorems. Both theorems lack a formal proof.

Second, Theorem 1 lacks proper definitions to make for a formally sound theorem statement. Further, the proof in the appendix doesn’t actually prove the theorem, or at least I cannot deduce how the proof proves the theorem statement. In particular, in the proof statement the authors do not define what it means for a sequence stored in KV cache to “exhibit translation-invariant properties”. Further, this requirement is not used in the proof. Further, the theorem states that the number KV cache items $C$ is upper bounded by some term. However, the proof does not actually arrive at this upper bound. There is a similar looking statement in the proof but I cannot see how this either proves the statement or whether it is even related to the upper-bound. Moreover, the bound uses the variables $b_k$ and $b_v$, but these variables are not defined anywhere. Finally, a note on the quality of the proofs (both the upper-bound and the equivalence to the delta update): These proofs are impossible for me to properly follow. The authors either work with equations that have never been introduced (e.g., in A.1.2, the authors state the update rule of a unique-filtered sequence but they neither define what they mean by update rule, nor what the update rule of a unique-filtered sequence is supposed to denote), use variables that are not defined (e.g., what is $t$ in A.1.1?), or introduce new notation on the fly ($S^{(KV)}$ in the last equation in A.1.2). In the current state, this cannot go through as a formal proof.

Third, Theorem 3 has a similar issue to Theorem 1, in that the proof statement and the proof are not compatible. The authors derive an approximation to attention in the proof but then they do not show the approximation error stated in the proof statement. Apart from that, I have no idea what $base\frac{2j}{d}$ is supposed to be. I do not understand what $base$ is supposed to be, as it is not defined anywhere, nor what $j$ and $d$ denote, as they are not defined anywhere. A similar issue arises with “the compressed positional information as a linear superposition”, which is not formally introduced. I can also not deduce what the authors mean here.
Minor note: The main text refers the reader to the wrong appendix section for the proof of Theorem 3.
The actual proof of Theorem 3 has similar issues than the proof of Theorem 1. In particular, I am missing a formal derivation or reference to the statement in Equation 26, a definition of $base$, a reference to what CRG NTK refers to, and a derivation of the approximation error to attention, as stated in the theorem statement. Again, this cannot go through as a formal proof.

Finally, in Theorem 5, it is not formally defined what multi-level decomposition means, and what “level” refers to in this context. Since this method is not formally defined, following the proof becomes difficult. Moreover, the proof uses some assumptions that are not mentioned in the theorem statement, for example that vocabulary vectors are linearly independent. Apart from this, I would say the proof of Theorem 5 is in the best condition among the five theorems, and the proof is also more closely aligned with the theorem statement.

**Requested Changes:**

In order to secure my recommendation for this work, the authors need to drastically improve the formal rigor in their mathematical analysis. In particular,

1. All used variables and concepts need to be *formally* defined.
2. A proof has to exactly prove the theorem statement and the authors need to ensure that the theorem statement logically follows from the proof.
3. Missing references need to be added and all assumptions used in the proof must be explicitly stated in the theorem statement.
4. Missing proofs (those which currently just have a proof sketch) need to be formalized and proved formally.

---

> ### Author Response · Authors · 2026-03-21
>
> Thank you again for your detailed and valuable feedback. Based on your comments, we have significantly revised the theoretical analysis, making it much clearer and more rigorous. Following the suggestion of another reviewer, we have also renamed the “Theorems” as “Propositions”. We will provide a summary response to your comments below, while the detailed theoretical revisions have been incorporated into the main text, which we believe will be clearer and easier to review than a lengthy reply.
>
> ---
>
> **First, for Theorems 2 and 4 the authors provide merely a proof sketch. This is clearly not enough to prove the theorems. Both theorems lack a formal proof.**
>
> *Response:*
> We have thoroughly revised Proposition 2 (formerly Theorem 2) by adding detailed explanations and reasoning to ensure that all concepts are clearly defined and the proof is rigorous. The main improvements include:
>
> - Clarifying all symbols: vocabulary size \(V\), sequence length \(T\), dimension \(d\), bit width \(b\), and the positional encoding function \(\text{PE}\).
> - Explicitly defining the meaning of “lossless compression” in the proposition statement and specifying the structure of the compressed representation \((\mathbf{E},\mathbf{I},\tilde{\mathbf{P}})\).
> - Structuring the proof into three parts: construction of the compressed representation, proof of lossless reconstruction, and derivation of the upper bound on the number of distinct vectors, followed by storage size analysis.
> - Removing discussion unrelated to the proposition (e.g., patterns in subsequent layers, connections to Proposition 3) to keep the proof focused.
> - Adding necessary assumptions (e.g., that word embeddings are injective) explicitly in the proposition statement.
>
> For Proposition 4 (formerly Theorem 4), we have completely rewritten it with clearly defined symbols and a rigorous proof. The core idea is to use numerical precision to guarantee a one-to-one correspondence between vectors with high similarity, thereby deriving a recursive upper bound.
>
> Detailed revisions have been made in the main text for your review.
>
> ---
>
> **Second, Theorem 1 lacks proper definitions to make for a formally sound theorem statement. Further, the proof in the appendix doesn’t actually prove the theorem, or at least I cannot deduce how the proof proves the theorem statement. In particular, in the proof statement the authors do not define what it means for a sequence stored in KV cache to “exhibit translation-invariant properties”. Further, this requirement is not used in the proof. Further, the theorem states that the number KV cache items \(C\) is upper bounded by some term. However, the proof does not actually arrive at this upper bound. There is a similar looking statement in the proof but I cannot see how this either proves the statement or whether it is even related to the upper-bound. Moreover, the bound uses the variables \(b_k\) and \(b_v\), but these variables are not defined anywhere. Finally, a note on the quality of the proofs (both the upper-bound and the equivalence to the delta update)......**
>
> *Response:*
> We have completely revised Proposition 1 (formerly Theorem 1) to ensure clarity of definitions, consistency of notation, logical rigor, and strict alignment between the proposition statement and its proof. The main modifications include:
>
> - Removing the undefined condition “translation-invariant properties.” Originally, we intended to convey that when decoding up to the current time step, the sequence of previous positions (after incorporating positional information) does not affect the output regardless of their order. Since this condition was not defined and did not affect the proof, we have removed it to avoid confusion.
> - Precisely defining all variables: bit width \(b\), key dimension \(d_k\), value dimension \(d_v\), count bit width \(b_c\), and \(C\) as the upper bound on the number of distinct sequence types.
> - Clearly stating the storage upper bound in the proposition and explaining what lossless compression means (equivalence of attention computation).
> - Structuring the proof into two parts: (1) derivation of the upper bound on the number of unique vectors; (2) proof of lossless attention equivalence via summation rearrangement.
> - Moving the discussion of equivalence to the Delta Rule to a separate corollary. For the update rule and related symbols that you pointed out, we have now provided clear definitions: the update rule follows the linear attention state update equations in Section 2.3, and the variable \(t\) denotes the time step in the sequence. All symbols are now defined in advance, and relevant references are cited.
>
> Detailed revisions have been made in the main text for your review.

---

> > ### Author Response · Authors · 2026-03-21
> >
> > ---
> >
> > **Third, Theorem 3 has a similar issue to Theorem 1, in that the proof statement and the proof are not compatible. The authors derive an approximation to attention in the proof but then they do not show the approximation error stated in the proof statement. Apart from that, I have no idea what \(base\frac{2j}{d}\) is supposed to be. I do not understand what \(base\) is supposed to be, as it is not defined anywhere, nor what \(j\) and \(d\) denote, as they are not defined anywhere. A similar issue arises with “the compressed positional information as a linear superposition”, which is not formally introduced. I can also not deduce what the authors mean here. Minor note: The main text refers the reader to the wrong appendix section for the proof of Theorem 3. The actual proof of Theorem 3 has similar issues than the proof of Theorem 1. In particular, I am missing a formal derivation or reference to the statement in Equation 26, a definition of \(base\), a reference to what CRG NTK refers to, and a derivation of the approximation error to attention, as stated in the theorem statement. Again, this cannot go through as a formal proof.**
> >
> > *Response:*
> > We have thoroughly revised Proposition 3 (formerly Theorem 3) to ensure that all variables and concepts are clearly defined, the proof is rigorous and stepwise, and it strictly aligns with the proposition statement. The main improvements include:
> >
> > - Clearly defining RoPE positional encoding, rotation angles \(\theta_j\), vector norms, etc.
> > - Providing a clear approximation formula and error bound in the proposition statement, with all variables (e.g., base, \(d\), \(j\), \(n\), \(s\)) explicitly defined.
> > - Structuring the proof into four steps: (1) expressing the RoPE inner product with rotated vectors; (2) applying small-angle approximations to \(\cos\) and \(\sin\) for large base; (3) performing a Taylor expansion of the exponential attention score and bounding the remainder; (4) deriving the final error bound.
> > - Introducing necessary assumptions (e.g., bounded vector norms, sufficiently large base to ensure small-angle approximations) explicitly in the proposition statement.
> > - Correcting unreasonable expressions in the original proposition (e.g., the undefined \(t\)) and replacing them with proper first-order approximations.
> >
> > Detailed revisions have been made in the main text for your review.
> >
> > ---
> >
> > **Finally, in Theorem 5, it is not formally defined what multi-level decomposition means, and what “level” refers to in this context. Since this method is not formally defined, following the proof becomes difficult. Moreover, the proof uses some assumptions that are not mentioned in the theorem statement, for example that vocabulary vectors are linearly independent. Apart from this, I would say the proof of Theorem 5 is in the best condition among the five theorems, and the proof is also more closely aligned with the theorem statement.**
> >
> > *Response:*
> > We have explicitly defined the concepts in Proposition 5 (formerly Theorem 5) and made minor revisions to the proof to ensure it is more rigorous, stepwise, and strictly aligned with the proposition statement. The main improvements include:
> >
> > - Clearly defining “level,” the various error terms, and the number of prototypes at each level.
> > - Simplifying the statement of the conclusions and ensuring that the proof directly corresponds to each conclusion.
> >
> > Detailed revisions have been made in the main text for your review.
> >
> > ---
> >
> > We sincerely thank you again for your insightful feedback, which has greatly helped us improve the clarity and rigor of our theoretical analysis. We hope that the revised manuscript now meets your expectations.

---

> > ### Comment · Reviewer_mgqY · 2026-04-20
> > **"Theorem" or now "Proposition" 1**
> >
> > your revision cleaned up the statements well, but i still don't see what this statement is for. do you use the number of bits S_b for anything?
> > basically, the only thing you state here is that number of unique kv pairs C is smaller than the total number of kv pairs T. the attention expression is commutative (which it is trivially by the virtue of being a sum), so instead of taking a sum over T terms Attention(k_t, v_t), you can rewrite it as a sum over the C unique pairs (k_i, v_i) weighted by their respective counts c_i.
> > the "necessity of deduplication" is motivated by the goal of compression, but not by any of the above facts.
> > overall, this "proposition" is still a heavy bloat for the expressed content. conversely, i find that the details about your method stated in the subsection "Generalization" ("when two tokens exhibit cosine similarity above a threshold, we treat them as identical and replace them by their average. This can be interpreted as a quantization process, which also provides noise reduction.") should be stated more prominently. however, while i agree that this is a quantization process, the statement "which also provides noise reduction" is not applicable: you are not reducing any noise, you are erasing information.

---

> > > ### Author Response · Authors · 2026-04-20
> > >
> > > Thank you very much for your further comments and questions. We truly appreciate your careful reading and thoughtful feedback, which are very helpful for improving our work.
> > >
> > > Regarding the use of $S_b$: In actual implementation, $S_b$ serves as a pre-defined upper bound on the total number of bits for the KV cache. However, we could also use $C$ instead of $S_b$ for both the statement of the proposition and the actual implementation. We will either add a discussion of the potential role of $S_b$ in the paper, or consider removing $S_b$ entirely if it is not used, as you suggested. Thank you very much for pointing this out.
> > >
> > > Regarding the purpose of stating the "necessity of deduplication" as a proposition: We did so to emphasize the fundamental fact of quantization in digital computers and the necessary efficient operations to achieve such a quantization upper bound in storage. Indeed, it is also motivated by compression, as you noted. However, we believe that this quantization and its associated upper bound are even more significant for our future work. Many of our architectures and training paradigms that aim to surpass standard attention and achieve what we consider genuine intelligence are based on the existence of such an upper bound, not merely on the goal of compression.
> > >
> > > Regarding the statement "which also provides noise reduction": We agree that this wording is inappropriate, as we have not provided experimental evidence or theoretical support for noise reduction. We will remove this statement. In future work, we may provide experimental support for such a claim, but for the current stage of this method, we cannot include relevant experiments, so we will delete it.
> > >
> > > Finally, thank you again for your thoughtful reading and valuable suggestions. We greatly appreciate your efforts to help us improve and refine our work. If you have any further questions, please do not hesitate to let us know. We would be very happy to continue the discussion. Thank you very much.

---

### Review · Reviewer_NdsN · 2026-03-22

**Summary Of Contributions:**

The paper introduces a framework called PLA that converts standard, memory-heavy AI models into more efficient "linear" models by smarter management of the data they store while processing text. The authors identify five specific mathematical principles, such as removing redundant information and compressing how the model remembers the order of words, to bridge the gap between complex and simple attention mechanisms. By using these principles, they created a model that can reuse parts of existing powerful AI like Llama or Mistral, making it much cheaper and faster to train. Their results show that this new approach performs as well as or better than other efficient models while using 20% less computing power during the fine-tuning process.

**Audience:**

Yes

**Audience Explanation:**

The paper is highly relevant to the TMLR audience, especially those working on machine learning theory and efficient models. It tackles an important problem: the high computation and memory cost of Transformers. The authors provide a clear mathematical connection between Softmax attention and linear models, which helps bring more understanding to an area that is often studied only through experiments. They also show how to convert pretrained models like Llama into linear versions, which is useful in practice. The error analysis is solid, and the proposed PLA model offers a good starting point for future work on KV cache compression and efficient inference.

**Broader Impact Concerns:**

I do not have broader impact concern on this paper.

**Claims And Evidence:**

Yes

**Claims Explanation:**

The submission presents a compelling combination of theoretical analysis and empirical validation. The authors introduce five principles for KV cache compression, each supported by clear mathematical derivations and associated error bounds. These principles are not merely theoretical; they are concretely instantiated in the design of PLA, establishing a coherent and well-motivated link between theory and implementation.


On the experimental side, the paper evaluates PLA on standard benchmarks as well as long-context tasks, where it consistently matches or surpasses prior linear models such as GSA and MVA. The results on resource efficiency are also notable, with the model achieving competitive performance while requiring only 80% of the usual fine-tuning compute. Additionally, the ablation studies are well-designed, isolating the contribution of each component and providing clear, comprehensive support for the overall approach.

**Requested Changes:**

1. The paper would benefit from a clearer breakdown of the “80% fine-tuning resource” claim, including which parts of the computation are actually reduced.
2. It would be helpful to compare PLA with more recent linear attention models, such as Mamba-2 or RWKV-6, to better show its position among current methods. Including real inference time (wall-clock time), not just theoretical complexity, would also make the efficiency claims more convincing.
3. The authors should discuss the limitations of inheriting weights, especially how much performance drops when moving from Softmax to PLA before fine-tuning.
4. Showing how the error changes with different sequence lengths would make it easier to see when compression starts to break down. A simpler explanation of the multi-level decomposition would also help make the theory easier to follow.

---

> ### Author Response · Authors · 2026-04-13
>
> **1. The paper would benefit from a clearer breakdown of the "80% fine-tuning resource" claim, including which parts of the computation are actually reduced.**
>
> The reason our method saves fine-tuning resources is that our PLA approach achieves smaller approximation error relative to standard attention compared to other linear models. This is theoretically guaranteed. Our method provides a currently optimal linearization path for standard attention from the perspective of KV cache compression, with error analysis provided at each compression node. We map the five compression nodes to the core mechanisms of current state-of-the-art models:
>
> - **Proposition 1 (redundancy removal)** corresponds to current Delta Rule-based approaches.
> - **Propositions 2 and 3 (compression of positional information and the quantization property of the tokenizer)** correspond to gating mechanisms or sequence-level dynamic decay biases.
> - **Proposition 4 (inter-layer similarity)** introduces cross-layer redundancy by controlling the upper bound of KV cache or linear state size between the first and last layers.
> - **Multi-level decomposition** achieves exponentially reduced approximation error, or equivalently, exponential expansion of effective state capacity—representing the current optimal approach.
>
> Through this tighter approximation to standard attention, we achieve substantial reduction in fine-tuning resources.
>
> ---
>
> **2. It would be helpful to compare PLA with more recent linear attention models, such as Mamba-2 or RWKV-6, to better show its position among current methods. Including real inference time (wall-clock time), not just theoretical complexity, would also make the efficiency claims more convincing.**
>
> In our original paper, we already compared PLA with more recent methods such as GSA and MVA, and included memory usage, prefill time, and inference latency in Figure 2. Since Mamba-2 and RWKV-6 do not perform well when fine-tuned from existing LLMs (likely requiring distillation), we instead compare 340M models trained from scratch. To isolate the efficiency of the attention mechanism, we compare 4-layer models of PLA, RWKV-6, and Mamba-2. Results are shown below.
>
> | Model | LMB acc↑ | PIQA acc↑ | Hella. acc↑ | Wino. accₙ↑ | ARC-e acc↑ | ARC-c acc↑ | Avg acc↑ |
> | :--- | :--- | :--- | :--- | :--- | :--- | :--- | :--- |
> | PLA-2 x 32 | 30.06 | 65.28 | 37.69 | 50.98 | 56.05 | 27.03 | 44.52 |
> | PLA-2 x 64 | 31.45 | 65.52 | 38.04 | 51.68 | 56.80 | 27.85 | 45.22 |
> | RWKV6-256 | 29.82 | 65.40 | 37.10 | 50.56 | 55.56 | 26.19 | 44.11 |
> | Mamba2-256 | 30.41 | 65.28 | 37.52 | 50.26 | 56.24 | 27.03 | 44.46 |
>
> **Inference times (ms)**
>
> | Seq Len | PLA-2 x 64 | PLA-2 x 32 | RWKV6-256 | Mamba2-256 |
> | :--- | :--- | :--- | :--- | :--- |
> | 8,192 | 45.83 | 32.51 | 38.90 | 34.73 |
> | 16,384 | 84.91 | 71.95 | 77.37 | 73.04 |
> | 32,768 | 160.58 | 131.07 | 156.49 | 147.06 |
> | 65,536 | 369.82 | 282.05 | 364.26 | 320.67 |
>
> Mamba-2 and RWKV-6 use a state size of 256. PLA uses two levels with state sizes 64 and 32 respectively. Since PLA's effective state capacity grows exponentially with the number of levels, its effective capacity is \(64^2 = 4096\). Therefore, Mamba-2 and RWKV-6 would require a state size of 4096 to fairly compare performance and efficiency.
>
> ---
>
> **3. The authors should discuss the limitations of inheriting weights, especially how much performance drops when moving from Softmax to PLA before fine-tuning.**
>
> First, the limitation of weight inheritance is that it cannot fully realize the potential of PLA or other linear models. This is because the modeling distribution of PLA differs from that of standard attention. LLMs are pre-trained with standard attention on massive datasets; fine-tuning with less than 0.1% of the data cannot fully convert the model to PLA's modeling distribution. Instead, it can only achieve the intersection of the two distributions, while the potentially superior parts of PLA lie outside this intersection. However, when the modeling capacity of linear models such as PLA fully covers that of standard attention, this issue will become less severe, requiring only more fine-tuning data.
>
> We already show the performance drop at each PLA node in Tables 1–4 of the original paper. Changing the distribution requires fine-tuning to compensate for the distribution shift. Following your suggestion, we also add training-free performance drop experiments: (next page)

---

> > ### Author Response · Authors · 2026-04-13
> >
> > - **Only Theory 1, Theory 2**: No performance loss; only decoupling and separate storage of information.
> > - **Compression of positional information (Theory 3)** : Causes significant performance loss. By retaining first-order positional information, we preserve some performance. The loss is similar to that of RoPE replacement methods (e.g., YaRN, PI, NTK), as the positional encoding distribution is altered.
> > - **Theory 1 + Theory 2 + Theory 4 (w/ cscale=1.6)** : Retains most information, as it only limits the diversity of tokens in intermediate layers. Performance loss is small for short sequences (e.g., ARC) but larger for long, complex tasks (e.g., MMLU).
> > - **Full PLA (Theory 5)** : Performance drops to random inference levels, because the multi-level storage distribution differs completely from the trained standard attention distribution, and additional weights are introduced. This is analogous to sparse attention without a mechanism like SnapKV to select relevant subsequences.
> >
> > | Method (no fine-tuning) | Passkey (1K-8K) | ARC | MMLU | SAMSUM |
> > | :--- | :--- | :--- | :--- | :--- |
> > | Mistral-7B-v0.1 | 100.0 | 54.0 | 62.4 | 43.6 |
> > | Only Theory 1 ($t=1$) | 100.0 | 54.0 | 62.4 | 43.6 |
> > | Only Theory 1 ($t=0.95$) | 92.5 | 49.6 | 55.2 | 40.7 |
> > | Only Theory 2&3 ($t=1.0$, w/ depos) | 100.0 | 54.0 | 62.4 | 43.6 |
> > | Only Theory 2&3 ($t=1.0$, w/ de&cprpos) | 22.5 | 42.1 | 38.6 | 28.8 |
> > | Only Theory 4 (w/ cscale=1.6) | 60.0 | 50.5 | 42.4 | 35.1 |
> > | Only Theory 5 (PLA: MVA+VI) | 0.0 | 28.3 | 25.4 | 19.8 |
> >
> > ---
> >
> > **4. Showing how the error changes with different sequence lengths would make it easier to see when compression starts to break down. A simpler explanation of the multi-level decomposition would also help make the theory easier to follow.**
> >
> > First, the errors in Theory 1 and Theory 2 are independent of sequence length. For Theory 3, Theory 4, and the full PLA, fine-tuning on different lengths leads to small performance changes on short sequences but affects performance beyond the window length, as shown below.
> >
> > | Method (fine-tuning 1B) | FT: 4K |  |  | FT: 8K |  |  | FT: 16K |  |  |
> > | :--- | :--- | :--- | :--- | :--- | :--- | :--- | :--- | :--- | :--- |
> > |  | Passkey (1K-4K) | (4K-8K) | (8K-16K) | (1K-4K) | (4K-8K) | (8K-16K) | (1K-8K) | (8K-16K) | (16K-32K) |
> > | Mistral-7B-v0.1 | 100.0 | 10.0 | 0.0 | 100.0 | 100.0 | 5.0 | 100.0 | 90.0 | 5.0 |
> > | Theory 3 ($t=1.0$, w/ de&cprpos) | 100.0 | 15.0 | 10.0 | 100.0 | 100.0 | 15.0 | 100.0 | 95.0 | 15.0 |
> > | Theory 4 (w/ cscale=1.6) | 100.0 | 10.0 | 10.0 | 100.0 | 100.0 | 10.0 | 100.0 | 100.0 | 10.0 |
> > | PLA (fine-tuning 8B) | 100.0 | 80.0 | 50.0 | 100.0 | 100.0 | 70.0 | 100.0 | 95.0 | 65.0 |
> >
> > **A simpler explanation of multi-level decomposition:**
> >
> > The memory storage and retrieval process of certain individuals—such as the author—follows a hierarchical pattern from concrete to abstract, and from coarse to fine. This is the inspiration for multi-level decomposition.
> >
> > **First level: fast, coarse "first impression"**
> > When first encountering a new concept, one extracts salient high-level features for rapid classification. These coarse categories form the first-level memory state, enabling fast access and approximate reasoning. For example, we can instantly distinguish broad biological categories like "mammals" from "birds" without needing fine-grained details. This process coarsely compresses the entire sequence into a fixed-size state. Since there is inevitably approximation error between the compressed representation and the original sequence, it is natural to compute and further compress the residual information—this becomes the second level.
> >
> > **Second level: fine-grained "deep processing"**
> > For more complex tasks, single-level coarse representations are insufficient. To improve performance, more precise information is needed: the second-level state stores residual information relative to the shallow memory (i.e., fine-grained corrections), rather than independent full information. This "difference storage" strategy ensures both completeness and efficient organization. For example, distinguishing individuals within the same species relies on subtle differences encoded as residuals; reasoning about long-term personality traits may require progressively finer residual representations across multiple levels.
> >
> > **Recursive storage** : As shown in our paper or the cited MVA work, recursively storing residuals from the previous level achieves exponential reduction in approximation error, thereby achieving exponential effective state capacity with minimal state size.

---

### Decision · Action_Editor_jq7A · 2026-04-30

**Recommendation:** Reject

**Audience:**

Yes

**Audience Explanation:**

The paper considers compression of the KV cache in large language models and proposes a method to replace softmax with linear attention, thereby overcoming the quadratic complexity of transformers. This touches on several interesting topics, LLM inference, linear attention, etc that are relevant to parts of the TMLR audience.

**Claims And Evidence:**

No

**Claims Explanation:**

Overall, the reviewers appreciated the empirical results of the paper, which demonstrate the efficiency of the proposed components. However, all reviewers expressed several concerns about the theoretical analysis. While some issues were addressed during the rebuttal, the presentation is still not sufficiently convincing and is, in parts, potentially misleading to the reader.

For example, Reviewer mgqY pointed out that the proposition in Subsection 3.1 is rather trivial, and that the information-theoretic framing may obscure rather than provide meaningful insights.

**Resubmission Of Major Revision:**

The authors may consider submitting a major revision at a later time.

---

> ### Author Response · Authors · 2026-05-03
>
> Thank you very much for your careful reading of our rebuttal and our paper, and for your timely decision. While we fully understand that the outcome is unlikely to change, we would like to offer a brief clarification regarding what we believe is a misunderstanding of our method.
>
> First, although the idea behind the proposition in Subsection 3.1 is indeed very simple, its implications for the field of linear attention are profound. Specifically, this proposition shows that standard attention is a special case of linear attention. Through the simple operation described in this proposition, the unboundedly growing KV cache can be transformed into a bounded KV cache—i.e., a fixed-size state can be maintained. We believe the importance of this conclusion warrants presenting it as a standalone proposition. This is analogous to a fundamental axiom in a new field, or a seemingly simple theorem derived from an axiom through straightforward reasoning: while the statement may be simple, it can have deep and lasting impact, and may even become a core component of the field.
>
> Second, regarding the “several concerns about the theoretical analysis” you mentioned, we believe that the current version of our paper does not mislead readers, and that we have adequately addressed the reviewers’ concerns. If the reviewers still have further questions, we would respectfully ask you to encourage them to raise those questions. We are confident that any remaining issues can be resolved relatively easily. Moreover, we note that Reviewer mgqY did not explicitly request a “major revision”; their attitude in the discussion was constructive rather than raising difficult or unresolved concerns. If appropriate, we would be happy to engage in further discussion with Reviewer mgqY to better understand their perspective and address any remaining issues.
>
> Furthermore, we acknowledge that each of our propositions is relatively simple in isolation. However, their deeper significance lies in their combination: they form a coherent linearization pathway that we believe will have substantial impact. This is reminiscent of calculus, where most individual operations are based on simple arithmetic, yet the combination of differentiation and integration has become an indispensable tool across many scientific and engineering disciplines.
>
>
> Finally, we hope the above clarification addresses your concerns. We remain open to any further questions and would be very glad to continue the discussion. Thank you again for your valuable time and effort. In addition, for certain reasons, we will de‑anonymize this paper and will not resubmit it to TMLR. Therefore, we greatly appreciate the opportunity to use the remaining time to conclude our discussion with you and address any further questions you may have. Once again, thank you very much for your time and for your careful reading of our paper and our discussion.